# Training Vision-Language Process Reward Models for Test-Time Scaling in Multimodal Reasoning: Key Insights and Lessons Learned

## Abstract

Process Reward Models (PRMs) provide step-level supervision that improves the reliability of reasoning in large language models. While PRMs have been extensively studied in text-based domains, their extension to Vision Language Models (VLMs) remains limited. Existing Vision-Language PRMs (VL-PRMs) rely on Monte Carlo Tree Search (MCTS) for data construction, which can often produce noisy supervision signals and limit generalization across tasks. In this work, we aim to elucidate the design space of VL-PRMs by exploring diverse strategies for dataset construction, training, and test-time scaling. First, we introduce a hybrid data synthesis framework that combines MCTS with judgments from a strong VLM, producing more accurate step-level labels. Second, we propose perception-focused supervision, enabling our PRM to explicitly detect errors at the visual grounding stage of reasoning. Third, we systematically evaluate multiple test-time scaling strategies, showing that our PRMs can reliably guide VLMs toward more accurate solutions. Our experiments covering five diverse multimodal benchmarks (MMMU, PuzzleVQA, AlgoPuzzleVQA, MathVista, and MathVision) reveal several key insights: (i) VL-PRMs when used as Outcome Reward Models (ORMs) during test-time scaling (TTS) can outperform VL-PRM guided process step selection, (ii) smaller VL-PRMs can match or even surpass larger ones in detecting process errors, (iii) VL-PRMs uncover latent reasoning abilities in stronger VLM backbones, (iv) perception-level supervision leads to significant gains in test-time scaling, and (v) TTS performance of different policies improve on advanced math reasoning datasets despite not training VL-PRMs on such datasets. We hope our work will motivate further research and support the advancement of VLMs. Our code is available at `https://anonymous.4open.science/r/mmr-eval-9B1E/`

## 1 Introduction

Chain-of-thought (CoT) (Wei et al., 2022) prompting has opened up new possibilities for Large Language Models (LLMs). Prior research demonstrates that CoT-based training can substantially improve LLM performance. A central challenge lies in obtaining high-quality training data, often addressed through distillation, where teacher models generate step-by-step reasoning traces. This approach has proven effective in models like DeepSeek-R1.

In such settings, effective *process supervision* becomes crucial. Rather than verifying only the final answer, it provides guidance and correction at every intermediate step, thereby ensuring logical consistency and factual reliability throughout the reasoning chain. Distillation data can be enhanced with process-level annotations, where each reasoning step is labeled as correct or incorrect, enabling the training of *Process Reward Models* (PRMs), which serve as reward functions for reinforcement learning (RL) with policy LLMs.

Beyond training, PRMs offer two key benefits: (1) identifying reasoning errors and (2) enabling test-time scaling by guiding inference. This is particularly important because LLMs remain vulnerable to *hallucinations* and subtle logical mistakes in multi-step reasoning tasks, such as mathematical problem solving (Wang et al., 2023; Zheng et al., 2024). Traditional outcome-only verifiers (Outcome Reward Models) are limited, evaluating only the final answer and often missing intermediate errors

that compromise the reasoning trajectory (Wang et al., 2024). In contrast, PRMs can detect such errors and translate them into effective guidance. During inference, PRMs can be leveraged to filter out flawed reasoning paths and steer generation toward more coherent and accurate solutions (Lightman et al., 2023; Zhang et al., 2025b).

Although research on PRMs has made significant progress, most existing work has focused on text-based tasks such as mathematical reasoning. More recently, (Wang et al., 2025) introduced a PRM tailored to vision-language models (VLMs). Their dataset was constructed using Monte Carlo Tree Search (MCTS), where the quality of a reasoning step was scored by the ratio of successful versus unsuccessful solution paths emanating from that step. This MCTS-based scoring strategy has also been widely explored in the math-focused PRM research (Zhao et al., 2025).

Despite Wang et al. (2025)'s progress, several key research questions on VL-PRMs remain open:

- **Beyond MCTS for dataset construction.** Wang et al. (2025) relied primarily on MC-score-based dataset construction, which strongly influenced PRM performance. Our central question is whether superior performance can be achieved by using stronger VLMs to judge each reasoning step, and subsequently employing these judgments as supervision signals for training VL-PRMs.

- **Does VL-PRM size matter?** Can smaller VL-PRMs match the performance of larger ones?

- **Strategies for test-time scaling.** How can VL-PRMs be best leveraged during inference?

- **Generalization across reasoning tasks.** While Wang et al. (2025) evaluated primarily on math-heavy datasets requiring expert subject knowledge, other reasoning domains remain underexplored. In particular, abstract reasoning tasks highlight human strengths but expose persistent weaknesses in VLMs. We ask whether VL-PRMs can bridge this gap.

- **The role of structured reasoning.** Prior studies Chia et al. (2024) suggest that perception is a major bottleneck in multimodal reasoning. Can VL-PRMs be designed to capture perception errors explicitly, and would such process-level supervision improve test-time scaling performance?

In this work, we expand the VL-PRM design space, offering insights across the full pipeline—from dataset construction and training to evaluation on diverse reasoning tasks.

- **Training data source.** Wang et al. (2025) train their model using advanced math datasets such as GeoQA+ (Cao & Xiao, 2022), CLEVR-Math (Lindström & Abraham, 2022), Geometry3K (Lu et al., 2021), GEOS (Seo et al., 2015), GeomVerse (Kazemi et al., 2023), and Geo170K (Gao et al., 2025). They evaluate on similarly complex benchmarks (MathVista, Dynamath, Wemath), yielding improved error detection, but this is expected given the overlap between training and evaluation. In contrast, we test whether training solely on general VQA and abstract reasoning, without advanced math datasets, can still enhance mathematical reasoning under test-time scaling (TTS). Our results confirm this: models trained this way consistently improve both TTS performance and error detection on advanced benchmarks, suggesting that general VQA and abstract reasoning tasks strengthen logical error detection, which in turn benefits mathematical reasoning under TTS.

- **Hybrid step evaluation.** While we also use MCTS to construct data, we depart from relying solely on MC-score-based metrics as in Wang et al. (2025). Instead, we additionally leverage a strong multimodal LLM (o4-mini) to directly judge the correctness of each step. This allows us to investigate whether LLM-based judgments provide more reliable process supervision than MC-score-based labeling. We are publicly releasing both the datasets [1].

- **Test-time scaling strategies.** We evaluate three VL-PRM-based methods: (1) *Greedy*, which iteratively selects the best next step at each decoding stage based on VL-PRM scores; (2) *One-shot*, which scores $N$ complete solutions and chooses the highest-scoring one; and (3) *Step-score Aggregation*, which averages step-level scores to select the final solution.

- **Perception-focused supervision.** Prior work, such as Chia et al. (2024), has shown that VLMs often struggle at the perception stage of multimodal reasoning even before doing inductive and deductive reasoning. Motivated by this, we train our VL-PRM to explicitly evaluate perception steps, which substantially improves test-time scaling performance of policy VLMs.

- **Backbone sensitivity.** We observe that different VLM backbones show varying degrees of reasoning capability when guided by PRMs at test time. For instance, Qwen-2.5-VL-7B and Gemma3-12B achieve comparable performance on PuzzleVQA and AlgoPuzzleVQA under standard evaluation,

---

[1]

yet with PRM-driven test-time scaling, Gemma3-12B outperforms Qwen-2.5-VL-7B by 8–15%. This highlights the role of PRMs in uncovering latent reasoning abilities of VLMs.

- **Comprehensive evaluation.** In contrast to Wang et al. (2025), we evaluate our approach on five diverse multimodal benchmarks covering various types of reasoning: *MMMU*, which requires expert subject knowledge; *PuzzleVQA*, targeting abstract reasoning; *AlgoPuzzleVQA*, focused on algorithmic reasoning; and *MathVista & MathVision*, designed for mathematical problem solving.

- **VL-PRM sizes.** We experiment with two VL-PRMs of different scales—3B and 7B parameters—initialized from Qwen-2.5-VL-3B and Qwen-2.5-VL-7B, respectively. Our results show that the smaller 3B model achieves comparable test-time scaling performance to the 7B model and outperforms the 7B variant by 5% in detecting process errors.

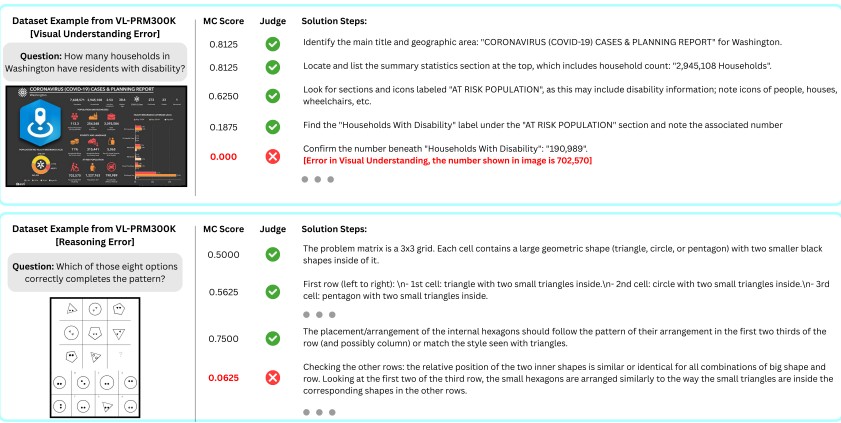

Figure 1: Visual and reasoning errors detected by MC scoring and o4-mini judge in VL-PRM300K.

## 2 METHOD

Data scarcity is a key challenge for training PRMs. While prior work uses MCTS-based scoring i.e., it assigns an *MC-score* to each reasoning step, e.g., $\text{MCScore}(i) = \frac{\text{\# correct completions from step } i}{\text{\# sampled completions from step } i}$ to build datasets like VISUALPRM400K (Wang et al., 2025), this dataset does not distinguish between perception and higher-level reasoning steps. To address this, we create a new dataset, VL-PRM300K, with structured traces that explicitly separate these stages. Additionally, we exclude advanced math datasets from training to study their impact on the generalizability of VL-PRMs across diverse multimodal reasoning tasks.

### 2.1 CONSTRUCTING VL-PRM300K

Let $\mathcal{I} = \{I_1, I_2, \ldots\}$ denote the image set and $\mathcal{Q}$ the question set. For a pair $(I, q) \in \mathcal{I} \times \mathcal{Q}$, a policy VLM $\pi$ is prompted to produce a stepwise solution $S = (s_1, s_2, \ldots, s_T)$, where $s_i$ denotes the $i$-th reasoning step. Under MCTS, from any prefix $s_{\leq i} \triangleq (s_1, \ldots, s_i)$ we expand continuations by rolling out the policy until termination: $s_{>i} \sim \pi(\cdot \mid I, q, s_{\leq i})$. The quality of step $s_i$ can then be assessed either by (i) the MCTS-derived $\text{MCScore}(i)$ as above, or (ii) an external judge VLM.

**Structured Reasoning.** Prior work has identified perception as a key bottleneck (Chia et al., 2024) in multimodal models. In particular, several studies show [2] (Jia et al., 2024) that prompting models to first describe or explain the image before attempting problem-solving significantly improves reasoning performance, as the additional grounding helps anchor subsequent inference (Wu et al., 2025; Thawakar et al., 2025) [3]. However, errors introduced at the perception stage can propagate through the reasoning process, ultimately degrading overall accuracy. Motivated by this observation,

---

[2] https://cloud.google.com/vertex-ai/generative-ai/docs/multimodal/design-multimodal-prompts#tailor_the_models_response_to_input

[3] https://andrewmayne.com/2024/03/12/improving-gpt-4s-visual-reasoning-with-prompting/

we design our VL-PRMs to evaluate not only reasoning steps but also perception steps, enabling them to detect and penalize errors at both stages of the structured reasoning pipeline.

**Using *o4-mini* as a Judge.** For judge-based supervision, we query a judge VLM $J$ (here, *o4-mini*) with the tuple $(I, q, s_{<i}, s_i)$ to decide whether step $s_i$ is correct given its context. Concretely, $J$ emits a binary label $y_i \in \{correct, incorrect\}$ along with a brief rationale. We use these judge labels to provide process-level supervision signals for training VL-PRMs. Following how PRMs are used at test-time, once a step is labeled as negative, we discard all subsequent steps.

**Dataset Statistics.** Our dataset comprises approximately 300K $(I, q, S)$ pairs, with overall statistics reported in Table 1. We sampled from six datasets covering diverse visual question answering (VQA) skills: document and chart understanding (DVQA); OCR (InfoVQA); general commonsense knowledge (VQAv2); grade-school level science (AI2D); grade-school level math reasoning (CLEVR-Math); and abstract reasoning (RAVEN). During dataset construction, we generate 5 solutions per image–question pair using GPT-4.1, with temperature of 1.0 and 16384 maximum new tokens. We select traces that adhere strictly to our structured perception-reasoning prompt format shown in Section A.4, discarding all invalid traces. Following the MCTS methodology described above, we split the perception and reasoning step traces into standalone steps, and for every step, we sample 16 continuations with prior steps (if any) included as prefixes. Finally, we query a judge VLM $J$ (here, *o4-mini*) to determine whether each step is correct given its context. The resulting VL-PRM300K dataset comprises approximately 300K samples, containing 77% of samples with an incorrect step found in the reasoning trace and 23% of samples with all steps verified correct, reported in Table 9. On average, each solution contains 4.55 steps, yielding an estimate of 1.32M step-level samples. Among these steps, about 17% are incorrect steps, with full details in Table 10. To study the effect of label imbalance, we additionally construct a balanced variant of VL-PRM300K.

**Constructing VL-PRM300K-balanced.** As reported in Table 1, VL-PRM300K is highly imbalanced, with the majority of samples containing samples with an incorrect step found in the step-by-step reasoning trace. Among the samples in VL-PRM300K, the majority of steps are labeled as correct, consistent with the observations in Wang et al. (2025). Among the error labels, approximately 86% correspond to perception errors, while the remaining 14% correspond to reasoning errors. To construct a balanced dataset with equal proportions of correct and incorrect samples, we first aggregate all samples from all datasets, excluding RAVEN. Next, we sample all the correct samples from RAVEN, since RAVEN contains significantly more samples with an incorrect step in its reasoning trace than samples with

| Statistics | Number |
|---|---|
| Total Samples | 290K |
|   - DVQA | 16K |
|   - InfoVQA | 22K |
|   - VQAv2 | 7K |
|   - AI2D | 21K |
|   - CLEVR-Math | 25K |
|   - RAVEN | 199K |
| Incorrect Samples | 77% |
| Correct Samples | 23% |
| Total Steps | 1.32M |
|   - Correct Steps | 1.10M |
|   - Incorrect Steps | 0.22M |
|     - Perception Errors | 86% |
|     - Reasoning Error | 14% |
| Avg. Number of Steps per Solution | 4.55 |

Table 1: Statistics of VL-PRM300K.

verified correct steps. Finally, we randomly select samples with incorrect steps from RAVEN to match the total number of correct samples, achieving a balanced distribution of correct and incorrect training samples, as summarized in Table 11.

## 2.2 TRAINING VL-PRMS

**Training.** VL-PRMs are trained on VL-PRM300K to predict "+" or "–" as step-level judgments. Training minimizes the cross-entropy loss over these label tokens for every step. The input to the VL-PRM is $\{I, q, s_0, p\}$. Here, the prompt $p$ is the system prompt for VL-PRM.

**Best-of-$N$ Inference.** Our VL-PRMs are used as a critic model to judge the Chain-of-Thought (CoT) step-by-step responses generated by the policy model. At inference time, the Vision-Language Process Reward Model (VL-PRM) receives an image–question pair $(I, q)$ along with a partial reasoning trajectory $\{s_1, \ldots, s_{i-1}\}$, and assigns a score to the candidate step $s_i$. The response that receives the highest score is chosen. We investigate three different strategies for integrating VL-PRMs into test-time scaling (Figure 4):

- **Guided Greedy Search.** Given $(I, q)$, the policy $\pi$ samples $N$ candidate next steps $\{s_i^{(1)}, \ldots, s_i^{(N)}\}$. Each candidate step is scored by the VL-PRM as $\text{Score}(s_i^{(j)}) = P_\theta(+ \mid I, q, s_{\leq i})$, where $P_\theta$ is the VL-PRM's probability of the step being correct. The highest-scoring step is selected, and $\pi$ continues generation conditioned on that step.

- **One-shot Search.** Instead of evaluating individual steps, the policy $\pi$ generates $N$ complete candidate solutions $\{S^{(1)}, \ldots, S^{(N)}\}$. VL-PRM scores each full solution in a single pass: $\text{Score}(S^{(j)}) = P_\theta(+ \mid I, q, S^{(j)})$. This approach avoids per-step evaluations, making it computationally cheaper than guided greedy search.

- **Step-Score Aggregation.** The policy $\pi$ first produces full solutions $\{S^{(1)}, \ldots, S^{(N)}\}$, where each solution $S^{(j)} = (s_1^{(j)}, \ldots, s_T^{(j)})$. VL-PRM assigns a per-step probability $p_i^{(j)} = P_\theta(+ \mid I, q, s_{\leq i}^{(j)})$, which is converted into a weighted score: $\tilde{s}_i^{(j)} = 1 \cdot p_i^{(j)} + 0 \cdot (1 - p_i^{(j)}) = p_i^{(j)}$. The overall solution score is then computed by aggregating step-level scores, e.g., using the mean: $\text{Score}(S^{(j)}) = \frac{1}{T} \sum_{i=1}^{T} \tilde{s}_i^{(j)}$. VL-PRM selects the solution with the highest aggregated score.

The policy models are instructed to do structured reasoning, where they first explain the image and then perform reasoning to solve the problem.

## 2.3 Constructing PerceptionProcessBench

We use VisualProcessBench to evaluate our VL-PRMs' ability to detect step-level visual reasoning errors. However, assessing their performance on perception error detection requires a dataset with explicit correctness annotations for perception steps. To address this, we synthetically construct PerceptionProcessBench from PuzzleVQA. In PuzzleVQA, each problem includes ground-truth perception details. To generate negative examples, we prompt GPT-4.1 to minimally alter the original perception step by either subtly modifying a correct detail or introducing a small, plausible error relevant to the problem. We specifically instruct the model to make only a single, subtle change, ensuring the resulting error is challenging for the VL-PRM to detect. Larger or multiple changes would make the error too obvious. As the original perception details are created using templates, to include variability in the steps, we further rephrase them using GPT-4.1. The exact prompt used for generating these mutations is provided in the Appendix. For each problem, we create one negative example, resulting in a dataset of 1,000 positive and 1,000 negative perception steps.

## 3 Experiments and Results

### 3.1 Experimental Settings

**Backbones.** We fine-tune Qwen-2.5-VL-3B and Qwen-2.5-VL-7B on the VL-PRM300K dataset, resulting in two models: **Qwen-VL-PRM-3B** and **Qwen-VL-PRM-7B**.

**Evaluation Datasets.** To evaluate these models under test-time scaling, we select five benchmark datasets that probe distinct aspects of multimodal reasoning: (i) **MMMU**, which requires expert-level subject knowledge, (ii) **PuzzleVQA**, designed to assess abstract reasoning, (iii) **AlgoPuzzleVQA**, targeting algorithmic reasoning, and (iv) **MathVista & MathVision**, which evaluates mathematical problem-solving ability. For **MMMU**, we use the validation set containing 900 problems. For **PuzzleVQA**, we construct an evaluation set of 1,000 samples by selecting the first 50 puzzles from each of 20 problem types. For **AlgoPuzzleVQA**, we similarly sample the first 50 puzzles from each of 18 problem types. For **MathVista**, we adopt the test-mini split, which contains 1,000 problems. Finally, for **MathVision**, we use the test split, which has 3040 samples. On all five datasets, the policy model is instructed to do structured reasoning, where the model first explains the image, followed by induction and deduction.

To assess the capability of our VL-PRMs in fine-grained step correctness evaluation, we use **VisualProcessBench** (Wang et al., 2025). This benchmark consists of 2,866 problems with step-by-step solutions, comprising a total of 26,950 steps manually annotated as *correct*, *incorrect*, or *neutral*. The problems are drawn from five datasets, including MMMU and four math benchmarks: MathVerse, DynaMath, MathVision, and WeMath. Performance on this benchmark is reported using the macro-F1 score. We dropped *neutral* samples from the data following Wang et al. (2025).

Table 2: Performance comparison across model families on visual reasoning datasets with Visual Process Reward Models (QWEN-VL-PRM-3B and QWEN-VL-PRM-7B). Improvements are shown in green with gains over base models.

| Model | MMMU | PuzzleVQA | AlgoPuzzleVQA | MathVista | MathVision | Overall |
|---|---|---|---|---|---|---|
| *Commercial Models* | | | | | | |
| GPT-4o | 70.7 | 60.0 | 57.8 | 30.9 | 31.2 | 50.1 |
| o1 | 78.2 | 78.9 | 54.4 | 73.9 | 60.3 | 69.1 |
| o3 | 82.9 | 84.1 | 62.3 | 86.8 | – | – |
| *Qwen-2.5-VL Family* | | | | | | |
| Qwen-2.5-VL-3B | 51.7 | 34.5 | 25.7 | 60.0 | 21.2 | 38.6 |
| + QWEN-VL-PRM-7B | 53.7 (+2.0) | 44.9 (+10.5) | 28.3 (+2.6) | 64.1 (+4.1) | 21.8 (+0.6) | 42.6 (+4.0) |
| Qwen-2.5-VL-7B | 55.0 | 48.0 | 29.1 | 67.8 | 24.2 | 44.8 |
| + QWEN-VL-PRM-3B | 57.6 (+2.6) | 55.5 (+7.5) | 33.8 (+4.7) | 70.0 (+2.2) | 26.1 (+1.9) | 48.6 (+3.6) |
| + QWEN-VL-PRM-7B | 57.4 (+2.4) | 54.8 (+6.8) | 35.3 (+6.2) | 71.0 (+3.2) | 26.2 (+2.0) | 48.9 (+4.1) |
| Qwen-2.5-VL-32B | 66.0 | 46.2 | 26.9 | 76.9 | 36.7 | 50.5 |
| + QWEN-VL-PRM-3B | 67.0 (+1.0) | 67.1 (+20.8) | 41.6 (+14.7) | 77.7 (+0.8) | 40.5 (+3.8) | 58.7 (+8.2) |
| + QWEN-VL-PRM-7B | 67.6 (+1.6) | 66.8 (+20.6) | 44.2 (+17.3) | 78.3 (+1.4) | 40.1 (+3.2) | 59.4 (+8.9) |
| *Gemma3 Family* | | | | | | |
| Gemma3-12B | 57.6 | 45.0 | 29.1 | 58.9 | 28.1 | 43.7 |
| + QWEN-VL-PRM-3B | 60.4 (+2.8) | 57.7 (+12.7) | 39.7 (+10.6) | 60.3 (+1.4) | 33.8 (+5.7) | 50.4 (+6.7) |
| + QWEN-VL-PRM-7B | 60.2 (+2.6) | 59.0 (+12.0) | 41.1 (+4.5) | 63.3 (+4.4) | 33.9 (+5.8) | 51.5 (+7.8) |
| Gemma3-27B | 62.9 | 50.8 | 29.9 | 61.6 | 32.4 | 47.5 |
| + QWEN-VL-PRM-3B | 65.5 (+2.6) | 67.4 (+16.6) | 40.3 (+10.4) | 65.4 (+3.8) | 39.8 (+7.4) | 55.7 (+8.2) |
| + QWEN-VL-PRM-7B | 64.5 (+1.6) | 67.6 (+16.8) | 41.1 (+11.2) | 65.2 (+3.6) | 40.9 (+8.5) | 55.9 (+8.4) |

## 3.2 MAIN RESULTS

### 3.2.1 VL-PRM IMPROVES PERFORMANCE UNDER TEST-TIME SCALING

**Results on MMMU.** Both QWEN-VL-PRM-3B and QWEN-VL-PRM-7B consistently enhance the performance of all evaluated policies under test-time scaling. On the expert-knowledge–oriented dataset **MMMU**, however, the observed gains are relatively modest, with improvements averaging around 2.5% across the three policies. This suggests that test-time scaling may have a limited impact in tasks where reasoning heavily depends on specialized domain knowledge.

**Results on Abstract and Algorithmic Reasoning Datasets.** In contrast, for datasets requiring abstract, algorithmic, or mathematical reasoning, we observe substantial improvements. On **PuzzleVQA**, performance increases by 7% for Qwen-VL-2.5-7B, 12.7% for Gemma3-12B, and 16.6% for Gemma3-27B. Similarly, on **AlgoPuzzleVQA**, we find improvements of approximately 5% for Qwen-VL-2.5-7B and 11% for both Gemma3-12B and Gemma3-27B. These results highlight that test-time scaling is particularly effective for reasoning-intensive tasks beyond expert subject knowledge. In Table 7, we present the performance of VisualPRM on the two datasets. VisualPRM was trained primarily on math-focused datasets and did not include any abstract reasoning data. The results show consistent performance gains across both datasets, with the exception of Qwen-2.5-VL-7B, where the improvement on PuzzleVQA is limited to just 1%. While these gains are notable, VisualPRM still does not surpass the performance of our VL-PRMs. Overall, these findings highlight that training on a mixture of general VQA and math datasets can transfer effectively to other reasoning domains, such as abstract and algorithmic reasoning.

**Results on Math Datasets.** On MathVista and MathVision, we observe a moderate performance improvement of 3–5%, which is lower than the gains seen on PuzzleVQA and AlgoPuzzleVQA. This is likely due to the lower proportion of math-related data in our training set compared to the abundance of abstract reasoning problems from RAVEN. Interestingly, despite using a much larger collection of math-focused problems, Wang et al. (2025) report a similar level of improvement on MathVista and MathVision across their models. However, when we evaluate VisualPRM using Qwen-2.5-VL-7B/32B and Gemma3-12B/27B as policy models, we find that on MathVista, performance drops from 1-2 percentage points (see Table 7). MathVision performance improves, but the improvement is less than our VL-PRMs. In our case, we include only a few samples from CLEVER-Math, which primarily consists of elementary problems such as basic addition and subtraction. Despite this limited mathematical data, our approach achieves comparable improvements to those of Wang et al. (2025), whose dataset includes more complex mathematical problems, such as those from GeoQA+, Geo170K, Geometry3K, GEOS, GeomVerse, and CLEVR-Math as well. We hypothesize that the substantial number of abstract reasoning and general VQA tasks in our dataset enhances the

Table 3: Performance on VisualProcessBench. Improvements over Qwen-2.5-VL-7B shown in green. **Bold** indicates best performance per column.

| Model | MathVision | MathVerse | MMMU | DynaMath | WeMath | Overall |
|---|---|---|---|---|---|---|
| GPT-4o-Mini | 53.6 | 58.9 | 57.1 | 56.7 | 58.5 | 57.9 |
| GPT-4o | **56.3** | **60.2** | **59.7** | 59.0 | 63.3 | 60.3 |
| Qwen-2.5-VL-3B | 43.9 | 42.6 | 41.6 | 45.2 | 40.8 | 43.4 |
| Qwen-2.5-VL-7B | 53.1 | 51.8 | 47.8 | 51.3 | 54.2 | 51.0 |
| Qwen-2.5-VL-32B | 58.4 | 63.3 | 57.5 | 61.1 | 60.3 | 59.5 |
| QWEN-VL-PRM-3B | 59.6 (+6.5) | 60.6 (+8.8) | 59.4 (+11.6) | **64.5** (+13.2) | **64.7** (+10.5) | **61.9** (+10.9) |
| QWEN-VL-PRM-7B | 55.7 (+2.6) | 58.8 (+7.0) | 55.8 (+8.0) | 58.3 (+7.0) | 59.8 (+5.6) | 58.6 (+7.6) |

general logical reasoning ability of the VL-PRM, thereby improving TTS performance across diverse domains.

**Test-time Scaling Unleashes Hidden Reasoning Ability.**    We observe that certain models, particularly Qwen-2.5-VL-32B, Gemma3-12B and Gemma3-27B, exhibit substantial gains under test-time scaling despite performing comparably to Qwen-VL-2.5-7B in the absence of scaling on PuzzleVQA and AlgoPuzzleVQA. While their baseline performance is similar, these larger Gemma models benefit disproportionately from the guidance of VL-PRMs during inference. This finding is noteworthy, as it suggests that process reward models (PRMs) can *unlock latent reasoning capacity* that is not apparent under standard greedy decoding.

A plausible explanation is that larger models often encode more sophisticated reasoning patterns, but these capabilities may remain underutilized due to suboptimal decoding trajectories. By filtering and guiding intermediate steps, VL-PRMs help these models explore more reliable reasoning paths, thereby revealing their hidden potential.

### 3.2.2   PERFORMANCE ON VISUALPROCESSBENCH

VisualProcessBench is employed to evaluate the ability of VL-PRMs to assess step-level correctness. Given a problem and its step-by-step solution, the VL-PRM computes the probabilities of the "+" and "–" tokens for each step. A step is labeled as positive if $P(+) > P(-)$, and negative otherwise. As baselines, we use Qwen-2.5-VL-3B and Qwen-2.5-VL-7B directly to predict step correctness. Our results in Table 3 show that both VL-PRMs outperform their respective baselines. While QWEN-VL-PRM-7B achieves a good improvement of 7.6% in macro-F1 score, QWEN-VL-PRM-3B exhibits a substantial gain of 16%, highlighting the effectiveness of PRM training even for smaller backbones. Additionally, we also benchmark GPT-4o-Mini and GPT-4o. We find that QWEN-VL-PRM-3B outperforms GPT-4o-Mini and performs comparably to GPT-4o.

**Takeaways.**    Our VL-PRMs are trained only on an elementary math dataset with basic operations, yet the 3B variant performs competitively with proprietary models like GPT-4o on VisualProcessBench, which contains challenging math datasets. This suggests that training on general VQA and abstract reasoning tasks effectively generalizes to mathematical reasoning for error detection. Future work will explore incorporating advanced math datasets (Wang et al., 2025) to further enhance performance.

Table 4: F1 scores of our VL-PRMs on PERCEPTIONPROCESS-BENCH.

| PRM | F1 |
|---|---|
| Qwen-VL-PRM-3B | **66.8** |
| w/o perception | 33.3 |
| Qwen-VL-PRM-7B | **70.3** |
| w/o perception | 33.3 |
| Qwen2.5-VL-3B-Instruct | 46.8 |
| Qwen2.5-VL-7B-Instruct | 56.3 |

### 3.2.3   RESULTS ON PERCEPTIONPROCESSBENCH

The results on PERCEPTIONPROCESSBENCH in Table 4 show that incorporating perception steps during training significantly improves the ability of VL-PRMs to detect perception errors. Our VL-PRMs outperform the baseline Qwen2.5-VL models of comparable size by a margin of 14%–20%. In contrast, when trained without perception supervision, the VL-PRMs tend to predict all perception steps as correct. This occurs because the training data does not explicitly indicate which perception steps are incorrect. Moreover, since our data is structured such that all subsequent steps are marked as negative after a negative step is identified by `o4-mini`, the model implicitly treats all perception steps as positive—even though their labels are masked during training.

## 3.3 ANALYSES AND ABLATIONS

**Findings from Test-time Scaling Strategies.** As outlined in Section 2.2, we investigate three strategies for leveraging VL-PRMs during test-time scaling. While all the TTS techniques improve performance substantially compared to the baseline, our results show that Guided Greedy Search consistently performs the worst among the three. A likely explanation is that selecting the immediately highest-rewarded step does not necessarily lead to an optimal reasoning trajectory due to error propagation from earlier steps. In fact, for challenging tasks such as AlgoPuzzleVQA, we even observe that greedy VL-PRM-guided search can reduce performance below the baseline policy. With Step-score aggregation, we see improvements across the benchmarks using all the different policies. Notably, both the greedy search and step-scorer approaches provide negligible improvement for MMMU.

Surprisingly, although our VL-PRMs are trained to assign rewards at the step level, performance improves markedly when the model is provided with an entire solution sequence for scoring, as described in Section 2.2. This suggests that, despite being trained for local supervision, VL-PRMs generalize effectively to full-solution evaluation. Access to complete contextual information appears to enable more accurate and reliable reward assignment, thereby en-

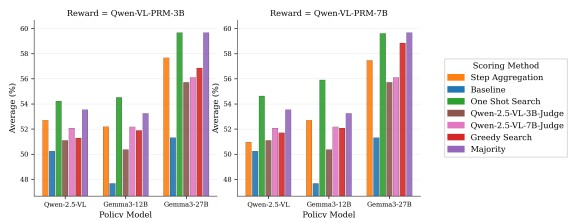

Figure 2: Comparison among different TTS methods.

hancing test-time scaling. The comparative results are reported in Figure 2. For Qwen-VL, although these two methods perform similarly on MMMU and MathVista, the differences are noticeable for PuzzleVQA and AlgoPuzzleVQA, where One-shot Search outperforms Guided Greedy Search by 4% to 9%. However, when looked closely, we find that Step-score aggregation tends to generate false positives, consequently affecting its performance. One reason for this could be the high imbalance of correct and incorrect steps in the training data.

One-shot Search has improved performance across VL-PRMs and benchmarks. This method is akin to Outcome Reward Modeling (ORM). What is interesting is that despite not training an explicit ORM, our PRM is acting as an ORM. This is in contrast to VisualPRM (Wang et al., 2025), where they trained an ORM explicitly, yet failed to perform as well as their PRM. To investigate this further, we evaluated VisualPRM on the same benchmarks and policy models, with details in Table 7. We found that VisualPRM shows improvement under the one-shot setting and only falls short of step-aggregation-based TTS by 1%. In our case, one-shot consistently outperforms step-aggregation by 2-3%.

**The Impact of Dataset Size.** Our experiments reveal that training on VL-PRM300K-balanced subset does not lead to consistent improvements: in some cases, performance increases marginally, while in others, a slight decrease is observed. We report this result Table 5.

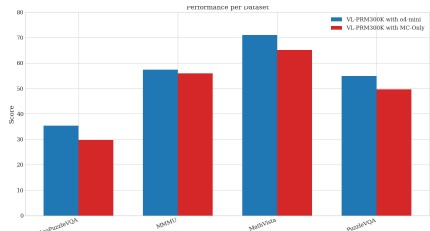

Figure 3: MC-Only vs o4-mini judge for VL-PRM300K construction.

**The Role of Judge Labels.** To construct VL-PRM300K, we first apply MCTS and then determine the correctness of each step using `o4-mini` as a judge. To assess whether an external judge LLM is necessary for improving VL-PRM performance, we experiment with MC-score–based labeling only to verify the correctness of steps. For this labeling scheme, as described in Section 2, a step is labeled as positive if the MC-score of a step exceeds a selected threshold, otherwise a step is labeled as negative. Our results using a threshold of 0, summarized in Figure 3, show that using `o4-mini` as the judge substantially improves VL-PRM performance compared to utilizing MC-score–based labeling only. We experiment with multiple thresholds, including 0.5 and 0.9, and find that using a judge continues to result in better overall performance. Furthermore, we measure the agreement between the two step-level labeling schemes and find that it is approximately approximately 20% for incorrect samples, underscoring the divergence between MCTS-derived supervision and LLM-based judgments.

Table 5: Performance comparison of VL-PRMs trained on variants of VL-PRM300K

| Policy Model | Reward Model | VL-PRM300K Variant | MMMU | PuzzleVQA | AlgoPuzzleVQA | MathVista | Average |
|---|---|---|---|---|---|---|---|
| Qwen-2.5-VL-7B | | *Baseline* | 55.0 | 48.0 | 29.1 | 67.8 | 49.98 |
| | Qwen-VL-PRM-3B | full | 55.33 | 55.50 | 33.78 | 69.00 | 53.40 |
| | | –balanced | 57.56 | 52.10 | 33.22 | 70.00 | 53.22 |
| | | –w/o perception | 54.22 | 49.10 | 33.11 | 67.90 | 51.08 |
| | Qwen-VL-PRM-7B | full | 57.44 | 54.80 | 33.78 | 69.80 | 53.96 |
| | | –balanced | 56.00 | 54.70 | 35.33 | 71.00 | 54.26 |
| | | –w/o perception | 53.89 | 51.80 | 32.11 | 68.20 | 51.50 |
| Qwen-2.5-VL-32B | | *Baseline* | 66.0 | 46.2 | 26.9 | 76.9 | 54.0 |
| | Qwen-VL-PRM-3B | full | 67.0 | 67.1 | 41.56 | 77.6 | 63.32 |
| | | –balanced | 65.78 | 67.0 | 40.33 | 77.4 | 62.63 |
| | | –w/o perception | 65.00 | 66.7 | 40.78 | 77.7 | 62.55 |
| | Qwen-VL-PRM-7B | full | 67.56 | 66.50 | 42.22 | 77.50 | 63.45 |
| | | –balanced | 65.00 | 66.80 | 44.22 | 78.30 | 63.58 |
| | | –w/o perception | 66.89 | 65.00 | 42.78 | 77.10 | 62.94 |
| Gemma3 12B | | *Baseline* | 57.6 | 45.0 | 29.1 | 58.9 | 47.65 |
| | Qwen-VL-PRM-3B | full | 60.00 | 57.70 | 39.67 | 60.30 | 54.42 |
| | | –balanced | 60.44 | 57.30 | 38.33 | 60.00 | 54.02 |
| | | –w/o perception | 57.44 | 54.80 | 36.11 | 57.30 | 51.41 |
| | Qwen-VL-PRM-7B | full | 59.11 | 59.00 | 40.78 | 63.30 | 55.55 |
| | | –balanced | 60.22 | 58.10 | 41.11 | 60.70 | 55.03 |
| | | –w/o perception | 58.44 | 55.60 | 37.67 | 58.20 | 52.48 |
| Gemma3 27B | | *Baseline* | 62.9 | 50.8 | 29.9 | 61.6 | 51.80 |
| | Qwen-VL-PRM-3B | full | 65.56 | 67.40 | 40.33 | 65.40 | 59.67 |
| | | –balanced | 65.11 | 67.20 | 39.33 | 64.70 | 59.09 |
| | | –w/o perception | 63.78 | 65.70 | 40.33 | 62.30 | 58.03 |
| | Qwen-VL-PRM-7B | full | 64.00 | 67.60 | 40.33 | 65.00 | 59.23 |
| | | –balanced | 64.56 | 67.00 | 41.11 | 65.20 | 59.47 |
| | | –w/o perception | 63.89 | 66.40 | 40.67 | 62.20 | 58.29 |

**Perception Guidance Enhances VL-PRMs.** During the construction of VL-PRM300K, we adopt a structured reasoning template in which the policy (o4-mini) first produces a perception-based description of the image, followed by inductive and deductive reasoning steps to solve the problem. This design enables training VL-PRMs on explicit perception steps. To evaluate the importance of perception supervision, we conduct an ablation study in which samples with perception errors are omitted during training. As shown in Table 5, excluding perception guidance leads to a consistent drop in performance for both QWEN-VL-PRM-3B and QWEN-VL-PRM-7B across all four benchmarks and with different policies. The performance drop is more evident in the 7B and 12B policy models compared to the 32B and 27B models, indicating that the larger models exhibit much stronger perception performance. Overall, these results demonstrate that perception guidance plays a crucial role in strengthening the capabilities of VL-PRMs.

**VL-PRMs vs. VLMs as Judges.** We evaluate whether VLMs can act as effective critics for test-time scaling (TTS). While using Qwen-2.5-VL models as judges in a one-shot best-of-$N$ setting improves performance, our VL-PRMs deliver substantially greater gains, consistently outperforming VLM-based judges. In contrast, when applying greedy search TTS with Qwen-2.5-VL-7B as both the policy and judge, performance drops below baseline across all datasets: MMMU (53.89%, ↓ 1%), PuzzleVQA (45.50%, ↓ 2.5%), AlgoPuzzleVQA (28.44%, ↓ 1.5%), and MathVista (65%, ↓ 3.1%). These results suggest that relying on VLMs for stepwise evaluation is not optimal.

**Majority Voting.** As shown in Figure 2, PRM-based test-time scaling outperforms majority voting for smaller models. Here, majority voting selects the final answer by choosing the most frequent output across multiple CoT traces. For larger models, such as Gemma 27B, the performance of PRM-based scaling and majority voting is comparable.

## 4 CONCLUSION

We introduce VL-PRM300K, a dataset built with MCTS and o4-mini labeling to train VL-PRMs for test-time scaling (TTS) of VLMs. Our study presents several in-depth and complementary analyses compared to the existing works. We show that when used as Outcome Reward Models (ORMs), VL-PRMs can outperform VL-PRM-guided response selection during TTS, with smaller models sometimes rivaling larger ones in error detection. They also reveal hidden reasoning abilities in strong VLM backbones, benefit greatly from perception-level supervision, and improve performance on advanced math tasks even without direct training on such datasets. These findings highlight the potential of VL-PRMs to drive progress in multimodal reasoning.

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

# A APPENDIX

## A.1 RELATED WORK

**Reward Models.** Reward models play a central role in reinforcement learning and test-time scaling by providing the feedback signal that guides model behavior (Christiano et al., 2023). Traditional Outcome Reward Models (ORMs) evaluate only the final output with a single score, whereas Process Reward Models (PRMs) assess intermediate steps in the reasoning trajectory and then aggregate these step-level evaluations into a final judgment. Recent multimodal PRMs, such as VisualPRM (Wang et al., 2025) and MM-PRM (Du et al., 2025) demonstrate improvement in reasoning tasks by constructing large process-supervision datasets and leveraging scalable step-level supervision. Follow-up works such as GM-PRM (Zhang et al., 2025a) and VRPRM (Chen et al., 2025) further refine error correction and visual reasoning capabilities. However, most existing multimodal PRMs focus on mathematical reasoning, with less focus on abstract visual reasoning (Zhang et al., 2019) in dataset construction. To address this gap, we introduce VL-PRM300K, the first multimodal process supervision dataset, which includes roughly 300K samples and 1.32 million steps with process supervision containing predominantly abstract reasoning samples, and develop VL-PRM, a multimodal PRM trained on this dataset.

**Test-time Scaling.** Most current multimodal PRM work only emphasizes on a step aggregation technique (Wang et al., 2025) with the focus mainly on the process reward model data construction. Some recent works further explore other TTS techniques, such as GM-PRM (Zhang et al., 2025a), which introduces a refined Best-of-N inference strategy in which the PRM not only scores multiple candidate reasoning paths but also corrects the first error in a path to guide further search. In ThinkPRM (Khalifa et al., 2025), it evaluates Best-of-N selection and reward-guided search across several math reasoning benchmarks. These works lack a comprehensive comparison of different test-time scaling (TTS) or inference time strategies. In our work, we evaluate a comprehensive range of TTS strategies, including greedy PRM-guided search, one-shot search, and step-wise score aggregation, to understand how they influence performance and generalization across various benchmarks.

## A.2 EXPERIMENTS

**TTS Strategies.** Different TTS strategies are shown in Figure 4. Detailed experimental results are shown in Table 6.

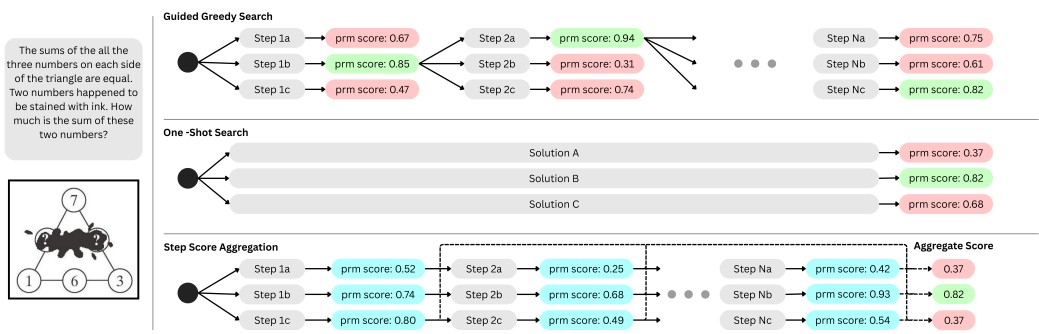

Figure 4: Different TTS strategies.

**Increasing N in BON@N.** In Figure 5, we see the effect of increasing N from 8 to 16 for Stepwise Score Aggregation and One-shot Search. The results suggest that performance improves by about 1% by increasing the size of N.

**Performance of VisualPRM.** We also evaluate VisualPRM on the same datasets and under both TTS settings used for our VL-PRMs. Unlike our models, VisualPRM does not show a consistent

Table 6: Performance comparison with color-coded results: **Best**, **Second**, and **Third** performing scaling methods per policy model and dataset. *Performance Ranking:* **Best** (Top 1), **Second** (Top 2), **Third** (Top 3) within each policy model per dataset. Gray background: Baseline (PRM-independent). Blue background: Majority (PRM-independent). The MathVision column is reserved for future evaluation results.

| Policy Model | Scoring Method | AlgoPuzzleVQA 3B | AlgoPuzzleVQA 7B | MMMU 3B | MMMU 7B | MathVista 3B | MathVista 7B | PuzzleVQA 3B | PuzzleVQA 7B | Average 3B | Average 7B |
|---|---|---|---|---|---|---|---|---|---|---|---|
| Qwen-2.5-VL-7B | Baseline | | 29.90 | | 54.95 | | 68.10 | | 48.00 | | 50.24 |
| | Step_agg | 29.78 | 27.67 | 55.62 | 54.39 | **69.70** | 67.60 | **55.70** | 54.20 | **52.70** | 50.97 |
| | One-shot | **33.80** | **35.30** | **57.60** | **57.40** | 70.00 | **71.00** | 55.50 | 54.80 | **54.23** | **54.63** |
| | Greedy | 27.89 | 27.95 | 57.34 | 56.56 | 67.90 | **71.20** | 52.00 | 51.10 | 51.28 | 51.70 |
| | Majority | | 31.44 | | 56.51 | | 73.20 | | 53.00 | | 53.54 |
| Gemma3 12B | Baseline | | 29.10 | | 57.67 | | 58.90 | | 45.00 | | 47.67 |
| | Step_agg | 33.89 | 36.78 | 57.67 | 57.00 | 61.40 | 61.30 | 55.80 | 55.70 | 52.19 | 52.70 |
| | One-shot | **39.67** | **41.11** | **60.44** | **60.22** | 60.30 | **63.30** | **57.70** | **59.00** | **54.53** | **55.91** |
| | Greedy | **36.89** | 38.22 | 55.56 | 54.67 | **62.40** | 61.20 | 52.70 | 54.20 | 51.89 | 52.07 |
| | Majority | | 37.11 | | 55.66 | | 61.60 | | 58.30 | | 53.17 |
| Gemma3 27B | Baseline | | 29.90 | | 62.95 | | 61.60 | | 50.80 | | 51.31 |
| | Step_agg | 38.11 | 38.78 | 61.89 | 62.11 | 65.00 | 63.90 | 65.70 | 65.10 | 57.68 | 57.47 |
| | One-shot | 40.33 | **41.11** | **65.56** | **64.56** | 65.40 | 65.20 | **67.40** | **67.60** | **59.67** | **59.62** |
| | Greedy | 38.00 | 40.00 | 60.22 | 62.33 | 63.10 | 64.70 | 66.10 | **68.40** | 56.86 | 58.86 |
| | Majority | | 42.00 | | 63.11 | | 63.80 | | 69.80 | | 59.68 |

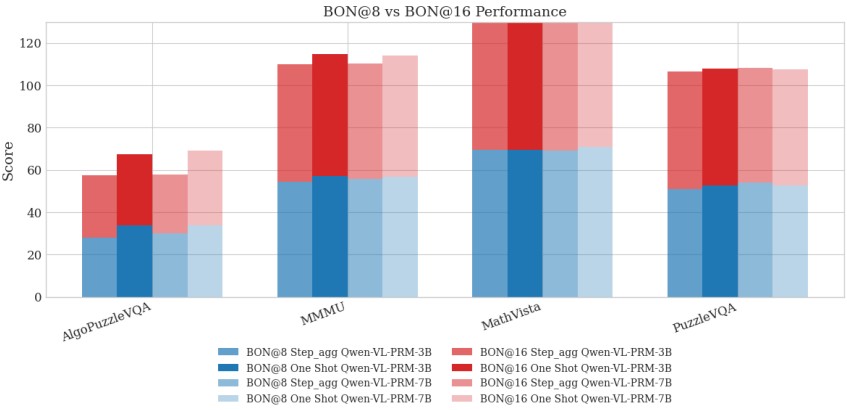

Figure 5: Comparison between BON@8 and BON@16 across datasets and models.

advantage for either Step Aggregation or One-shot TTS. For instance, with Qwen-2.5-VL-7B, One-shot search outperforms Step Aggregation by only 0.6%, while for other models Step Aggregation has a slight edge of about 1% on average across datasets.

Although VisualPRM was primarily trained on math-focused datasets, it performs reasonably well on abstract and algorithmic reasoning tasks, demonstrating some level of generalization. However, its overall performance gains are notably smaller than those of our VL-PRMs. This gap likely arises from the higher proportion of abstract reasoning samples in VL-PRM300K, which we use to train our VL-PRMs.

We expected VisualPRM to achieve its strongest improvements on math datasets such as *MathVista* and *MathVision*. While it shows consistent gains on *MathVision*, performance on *MathVista* decreases by 1–2 percentage points. Even on *MathVision*, the gains do not surpass those of our VL-PRMs. Several factors may contribute to this difference, including variations in model backbones, differences in dataset composition, and the use of GPT-4.1 for generation and o4-mini for judgment used to train our models instead of using InternVL-2.5-7B for generation and MCScore threshold-based step labeling only used for training VisualPRM.

Interestingly, on MMMU, VisualPRM exhibits a performance decline across nearly all settings, with the sole exception of Qwen-2.5-VL-7B, where it achieves a modest 2% improvement.

Table 7: Performance comparison across model families using VisualPRM (Wang et al., 2025). Improvements are shown in green with gains over base models.

| Model | MMMU | PuzzleVQA | AlgoPuzzleVQA | MathVista | MathVision | Overall |
|---|---|---|---|---|---|---|
| ***Commercial Models*** | | | | | | |
| GPT-4o | 70.7 | 60.0 | 57.8 | 30.9 | 31.2 | 50.1 |
| o1 | 78.2 | 78.9 | 54.4 | 73.9 | 60.3 | 69.1 |
| o3 | 82.9 | 84.1 | 62.3 | 86.8 | – | – |
| ***Qwen-2.5-VL Family*** | | | | | | |
| Qwen-2.5-VL-3B | 51.7 | 34.5 | 25.7 | 60.0 | 21.2 | 38.6 |
| + Qwen-VL-PRM-7B | 53.7 (+2.0) | 44.9 (+10.5) | 28.3 (+2.6) | 64.1 (+4.1) | 21.8 (+0.6) | 42.6 (+4.0) |
| Qwen-2.5-VL-7B | 55.0 | 48.0 | 29.1 | 67.8 | 24.2 | 44.8 |
| + VisualPRM-Step-Agg | 57.1 (+2.1) | 49.1 (+1.1) | 27.7 (-1.4) | 68.1 (+0.3) | 25.7 (+1.5) | 45.5 (+0.7) |
| + VisualPRM-One-shot | 55.7 (+0.7) | 48.0 (+0.0) | 33.5 (+4.4) | 66.0 (-1.8) | 27.6 (+3.4) | 46.1 (+1.3) |
| + Qwen-VL-PRM-3B | 57.6 (+2.6) | 55.5 (+7.5) | 33.8 (+4.7) | 70.0 (+2.2) | 26.1 (+1.9) | 48.6 (+3.6) |
| + Qwen-VL-PRM-7B | 57.4 (+2.4) | 54.8 (+6.8) | 35.3 (+6.2) | 71.0 (+3.2) | 26.2 (+2.0) | 48.9 (+4.1) |
| Qwen-2.5-VL-32B | 66.0 | 46.2 | 26.9 | 76.9 | 36.7 | 50.5 |
| + VisualPRM-Step-Agg | 62.4 (-3.6) | 62.9 (+16.7) | 42.4 (+15.5) | 76.7 (-0.2) | 39.5 (+2.8) | 56.7 (+6.2) |
| + VisualPRM-One-shot | 61.8 (-4.8) | 60.0 (+13.8) | 40.7 (+13.8) | 75.4 (-1.5) | 39.5 (+2.8) | 55.5 (+5.0) |
| + Qwen-VL-PRM-3B | 67.0 (+1.0) | 67.1 (+20.8) | 41.6 (+14.7) | 77.7 (+0.8) | 40.1 (+3.4) | 58.7 (+8.2) |
| + Qwen-VL-PRM-7B | 67.6 (+1.6) | 66.8 (+20.6) | 44.2 (+17.3) | 78.3 (+1.4) | 40.4 (+3.7) | 59.4 (+8.9) |
| ***Gemma3 Family*** | | | | | | |
| Gemma3-12B | 57.6 | 45.0 | 29.1 | 58.9 | 28.1 | 43.7 |
| + VisualPRM-Step-Agg | 57.1 (-0.5) | 52.8 (+7.8) | 33.0 (+3.9) | 57.8 (-1.1) | 28.1 (+0.0) | 46.9 (+3.2) |
| + VisualPRM-One-shot | 57.5 (-0.1) | 50.1 (+5.1) | 34.3 (+5.2) | 56.8 (-2.1) | 32.5 (+4.4) | 46.2 (+2.5) |
| + Qwen-VL-PRM-3B | 60.4 (+2.8) | 57.7 (+12.7) | 39.7 (+10.6) | 60.3 (+1.4) | 33.8 (+5.7) | 50.4 (+6.7) |
| + Qwen-VL-PRM-7B | 60.2 (+2.6) | 59.0 (+12.0) | 41.1 (+4.5) | 63.3 (+4.4) | 33.9 (+5.8) | 51.5 (+7.8) |
| Gemma3-27B | 62.9 | 50.8 | 29.9 | 61.6 | 32.4 | 47.5 |
| + VisualPRM-Step-Agg | 62.3 (-0.6) | 64.8 (+14.0) | 38.5 (+8.6) | 61.8 (+0.0) | 27.8 (-4.6) | 53.0 (+5.5) |
| + VisualPRM-One-shot | 61.1 (-1.1) | 62.4 (+11.6) | 39.6 (+9.7) | 59.6 (-2.0) | 37.2 (+4.8) | 52.0 (+4.5) |
| + Qwen-VL-PRM-3B | 65.5 (+2.6) | 67.4 (+16.6) | 40.3 (+10.4) | 65.4 (+3.8) | 39.8 (+7.4) | 55.7 (+8.2) |
| + Qwen-VL-PRM-7B | 64.5 (+1.6) | 67.6 (+16.8) | 41.1 (+11.2) | 65.2 (+3.6) | 40.9 (+8.5) | 55.7 (+7.7) |

**Limited TTS benefits on MiniCPM-V family of models.** We observed limited test-time scaling benefits for openbmb/MiniCPM-V-2_6 across all benchmarks, as shown in table 8, averaging 2% improvement overall. Upon comparison of baseline model outputs (in the absence of test-time scaling) and the TTS temperature-sampled outputs generated, we noticed that the baseline model is more likely to arrive at final correct answers when generating step-by-step traces in Mandarin Chinese. Building on this insight, we investigated various methods to elicit the model to "think" in Mandarin Chinese in the hope of better TTS performance, but faced limited systematic performance improvements. Future work will explore how we can further enhance performance for such models. We also faced challenges reproducing the baseline scores reported for MathVision, despite following the configurations detailed in VLMEvalKit specifically for evaluating MiniCPM-V-2_6.

Table 8: MiniCPM-V-2.6 performance on visual reasoning datasets (when used with QWEN-VL-PRM-7B). Improvements are shown in green with gains over base models.

| Model | MMMU | PuzzleVQA | AlgoPuzzleVQA | MathVista | MathVision | Overall |
|---|---|---|---|---|---|---|
| *MiniCPM-V Family* | | | | | | |
| MiniCPM-V-2.6 | 47.8 | 36.0 | 27.9 | 60.8 | 18.1* | 38.1 |
| + QWEN-VL-PRM-7B | 50.0 (+2.2) | 37.9 (+1.9) | 30.2 (+2.3) | 62.2 (+1.4) | 20.1* (+2.0) | 40.1 (+2.0) |

## A.3 DATASET ANALYSIS

**Details** In this Section, we will provide details on VL-PRM300K.

Table 9: Breakdown of VL-PRM300K by data source.

| Source | RAVEN | CLEVR-Math | InfoVQA | VQAv2 | AI2D | DVQA | Overall |
|---|---|---|---|---|---|---|---|
| *Cumulative Statistics* | | | | | | | |
| Total Valid Samples | 203,115 | 27,326 | 25,518 | 8,138 | 22,399 | 16,663 | 303,159 |
| Total Samples Discarded | 4,610 | 1,849 | 3,768 | 1,010 | 1,428 | 488 | 13,153 |
| Total Training Samples | 198,505 | 25,477 | 21,750 | 7,128 | 20,971 | 16,175 | 290,006 |
| *Consensus Filtering Breakdown (number of rows)* | | | | | | | |
| o4-mini Incorrect Step Identified | 197,028 | 10,134 | 3,828 | 2,478 | 6,614 | 4,233 | 224,315 |
| o4-mini Correct and MC Consensus | 1,477 | 15,343 | 17,922 | 4,650 | 14,357 | 11,942 | 65,691 |
| o4-mini Correct and MC Disagree | 4,610 | 1,849 | 3,768 | 1,010 | 1,428 | 488 | 13,153 |
| *Breakdown of steps with Perception v.s. Reasoning Errors (%)* | | | | | | | |
| Perception Incorrect (%) | 86.4 | 96.3 | 81.5 | 59.1 | 80.6 | 70.4 | - |
| Reasoning Incorrect (%) | 13.6 | 3.7 | 18.5 | 40.9 | 19.4 | 29.6 | - |

Table 10: Breakdown of steps in VL-PRM300K by data source.

| Source | RAVEN | CLEVR-Math | InfoVQA | VQAv2 | AI2D | DVQA | Overall |
|---|---|---|---|---|---|---|---|
| *Cumulative Statistics* | | | | | | | |
| Total Steps | 619,692 | 160,632 | 171,890 | 52,426 | 170,683 | 145,360 | 1,320,683 |
| *Number of Steps Breakdown* | | | | | | | |
| Incorrect Steps | 197,028 | 10,134 | 3,828 | 2,478 | 6,614 | 4,233 | 224,315 |
| Correct Steps | 422,664 | 150,498 | 168,062 | 49,948 | 164,069 | 141,127 | 1,096,368 |
| *Step Breakdown (%)* | | | | | | | |
| Incorrect Steps (%) | 31.8 | 6.3 | 2.2 | 4.7 | 3.9 | 2.9 | - |
| Correct Steps (%) | 68.2 | 93.7 | 97.8 | 95.3 | 96.1 | 97.1 | - |

Table 11: Breakdown of VL-PRM300K-balanced by data source

| Source | RAVEN | CLEVR-Math | InfoVQA | VQAv2 | AI2D | DVQA | Overall |
|---|---|---|---|---|---|---|---|
| *Cumulative Statistics* | | | | | | | |
| All Training Samples | 198,505 | 25,477 | 21,750 | 7,128 | 20,971 | 16,175 | 290,006 |
| Balanced Training Samples | 39,881 | 25,477 | 21,750 | 7,128 | 20,971 | 16,175 | 131,382 |
| *Number of rows* | | | | | | | |
| Samples with Incorrect Step | 38,404 | 10,134 | 3,828 | 2,478 | 6,614 | 4,233 | 65,691 |
| Samples with Verified Correct Steps | 1,477 | 15,343 | 17,922 | 4,650 | 14,357 | 11,942 | 65,691 |
| *Proportion of rows used in training (%)* | | | | | | | |
| Samples with Incorrect Step (%) | 96.3 | 39.8 | 17.6 | 34.8 | 31.5 | 26.2 | - |
| Samples with Verified Correct Steps (%) | 3.7 | 60.2 | 82.4 | 65.2 | 68.5 | 73.8 | - |

## A.4 PROMPTS

**Details** In this Section, we will provide details regarding the prompts used in VL-PRM300K construction and test time evaluation.

GPT-4.1 Rollout Prompt for RAVEN during Dataset Generation

*You are an abstract reasoning puzzle expert. The puzzle you will receive is presented in a standard Raven's Progressive Matrices format: a $3 \times 3$ matrix of related images, with the bottom-right cell (the ninth tile) missing. There are eight possible answer choices provided separately, and your task is to decide which of those eight images correctly completes the $3 \times 3$ matrix pattern.*

*I will provide you with an image containing:*

- ***Problem Matrix:*** *An accompanying image that shows the eight tiles and highlights where the ninth is missing.*

- ***Answer Set:*** *The eight candidate images from which you must choose the best fit for the missing tile.*

*Your task is to:*

- *Review the problem matrix and the accompanying image in sequence, describing step-by-step what you see in the image in* `<perception>` *tags.*

- *Reason step-by-step about the logical pattern or rule connecting the tiles in* `<reasoning>` *tags.*

- *Deduce the correct tile from the eight provided options in* `<correct_answer>` *tags.*

*It is crucial that your solution contains these sections in the exact format described below:*

```
[Perception]
<step_1>
...(Step 1 of step-by-step perception)...
</step_1>
<step_2>
...(Step 2 of step-by-step perception)...
</step_2>
...
<step_n>
...(Step n of step-by-step perception)...
</step_n>
[Reasoning]
<step_1>
...(Step 1 of step-by-step reasoning)...
</step_1>
<step_2>
...(Step 2 of step-by-step reasoning)...
</step_2>
...
<step_m>
...(Step m of step-by-step reasoning)...
</step_m>
<correct_answer>
...(Clearly state which of the 8 candidate images is the best
candidate image as the missing tile to complete the matrix. If
the candidates are numbered, lettered, or can be uniquely
described, use that identifier.)...
</correct_answer>
```

*Now proceed with the task.*

GPT-4.1 Rollout Prompt for AI2D during Dataset Generation

*You are an expert data analyst specializing in interpreting data visualizations. Your task is to answer questions about charts and graphs presented to you in images. You will be provided*

*with an image containing one or more data visualizations and a specific question about the data presented.*
*I will provide you with an image containing:*

- *Data Visualization: A chart or graph that contains data points and other visual elements.*

*Here's the question you need to answer:*

```
<question>
{{QUESTION}}
</question>
```

*Please follow these steps to complete the task:*

1. *Carefully examine the image, paying attention to all elements of the data visualization(s) such as titles, labels, axes, legends, and data points.*

2. *Analyze the data presented in the visualization(s), identifying specific data points or trends that relate to the question asked.*

3. *Interpret the data and connect it to the specific question asked. Consider how the data directly relates to answering the question.*

4. *Use your analysis and interpretation to determine the answer to the question. The answer must be a single integer.*

5. *Present your answer in a LaTeX-formatted box using this format:*

```
<correct_answer>
\boxed{Your answer here}
</correct_answer>
```

**Your task is to:**

- *Under the [Visual Elements] section, list out all relevant visual elements step-by-step that relate to answering the question. Be thorough but concise. Wrap each step in* `<step_1>, <step_2>, ...` *tags.*

- *Under the [Reasoning] section, explain your step-by-step reasoning process. This should include your analysis, interpretation, and how you arrived at the answer. Provide a clear justification of how you derived the answer from the data presented. Wrap each step in* `<step_1>, <step_2>, ...` *tags.*

- *Present your final answer using the LaTeX-formatted box in* `<correct_answer>` *tags.*

*It is crucial that your solution contains these sections in the exact format described below:*

```
[Visual Elements]
<step_1>
...(Step 1 of step-by-step perception)...
</step_1>
<step_2>
...(Step 2 of step-by-step perception)...
</step_2>
...
<step_n>
...(Step n of step-by-step perception)...
</step_n>

[Analysis and Interpretation]
<step_1>
...(Step 1 of step-by-step reasoning)...
</step_1>
<step_2>
```

```
...(Step 2 of step-by-step reasoning)...
</step_2>
...
<step_m>
...(Step m of step-by-step reasoning)...
</step_m>

<correct_answer>
$\boxed{\text{integer}}$
</correct_answer>
```

***Remember to***:

- *Provide only a single string answer in the* `<correct_answer>` *section using the* `$\boxed{string_answer}$` *format, and no other text or commentary.*

---

**Qwen Policy Prompt for TTS Genertion**

$< |im\_start| >$*system*
*You are an expert visual analyst who solves complex visual reasoning problems by systematically connecting what you observe in images to the specific requirements of each problem. First, identify and inventory all problem-relevant visual elements through a step-by-step perceptual process, documenting their spatial relationships, object identities, and perceptual details with precision at each stage of observation. Then, construct a step-by-step logical pathway that explicitly shows how each visual observation leads to your solution, explaining why each detail matters and citing specific image evidence at every step. Your analysis will be evaluated by a teacher on two criteria: \*\*Visual Elements\* (accuracy of perception, spatial understanding, and object recognition) and \*Reasoning\*\* (logical consistency, valid deductive steps, and sound analytical conclusions). Multiple solution pathways will be explored in parallel at each step, building upon previous reasoning to generate diverse approaches, with the teacher scoring each path to accept or reject based on visual accuracy and logical validity. This approach increases the odds of reaching a correct final answer by exploring and focusing on perception and reasoning traces that are valid and correct while ignoring incorrect ones. This makes your precise step-by-step perception and reasoning essential for identifying the most promising solution.$< |im\_end| >$*
$< |im\_start| >$*user*
$< |vision\_start| >< |image\_pad| >< |vision\_end| >$ *Question:* **{{Question}}** *Options:*
**{{Options}}**
*Please select the correct answer from the options above. If you are uncertain or the problem is too complex, make a reasoned guess based on the information provided. Avoid repeating steps indefinitely—provide your best guess even if unsure. Determine whether to think step by step based on the difficulty of the question, considering all relevant information before answering. If you do need to think step by step, first, carefully observe and describe everything you see: the image itself and how it connects to the problem being presented. Then, work through your reasoning step by step to arrive at your answer. \*Your teacher will review your descriptions of visual elements to ensure you're observing all relevant details accurately, and will critique each step of your reasoning to provide guidance and ensure you're on the right track.\* Put your final answer within □. If multiple-choice options for answers to the question are provided, select the alphabetical letter of the corresponding best option within* \boxed{} *when you are ready to provide your final answer.$< |im\_end| >$*
$< |im\_start| >$*assistant*

---

**o4-mini Judge Prompt for Sample Correctness Verification**

*I will provide an abstract visual reasoning problem along with a solution. They will be formatted as follows:*
*[Abstract Visual Reasoning Problem]*

```
<abstract_visual_reasoning_problem>
```

```
...(abstract visual reasoning problem)...
</abstract_visual_reasoning_problem>
```

*[Solution]*

```
<solution>
[Perception]
<step_1>
...(Step 1 of step-by-step perception)...
</step_1>
<step_2>
...(Step 2 of step-by-step perception)...
</step_2>
...
<step_m>
...(Step m of step-by-step perception)...
</step_m>
[Reasoning]
<step_1>
...(Step 1 of step-by-step reasoning)...
</step_1>
<step_2>
...(Step 2 of step-by-step reasoning)...
</step_2>
...
<step_n>
...(Step n of step-by-step reasoning)...
</step_n>
<correct_answer>
...(The provided answer to the abstract visual reasoning problem)...
</correct_answer>
</solution>
```

*Your task is to review each paragraph of the solution in sequence, analyzing, verifying, and critiquing the visual elements perception and reasoning in detail. You need to provide the analyses and the conclusion in the following format:*

*[Perception]*

```
<analysis_1>
...(analysis of step 1)...
</analysis_1>
...
<analysis_m>
...(analysis of step m)...
</analysis_m>
```

*[Reasoning]*

```
<analysis_1>
...(analysis of step 1)...
</analysis_1>
...
<analysis_n>
...(analysis of step n)...
</analysis_n>
<conclusion>
Correct/Incorrect
</conclusion>
```

- *When you analyze each step, you should use proper logical or perceptual verification as appropriate, or reflection to indicate whether it is logically and perceptually valid. Please carefully go through this process.*

- *Each analysis should:*
  - *Check if the described pattern/rule is actually present*
  - *Verify the logical consistency of the step*
  - *Confirm that conclusions follow from observations*
- *If an error is detected in any step, you should describe the nature and cause of the error in detail, and suggest how to correct the error or the correct approach.*
- *When the step is found to contain an error, stop further analysis of subsequent steps (as they may depend on the identified error) and directly provide the conclusion of "Incorrect" in the <conclusion> tag.*
- *Only material reasoning or perception errors should trigger the early termination of the analysis.*
- *For instance, given a solution of five steps, if an error is found in the third step, you should reply in the following format:*

```
<analysis_1>
...(analysis of step 1)...
</analysis_1>
<analysis_2>
...(analysis of step 2)...
</analysis_2>
<analysis_3>
...(analysis of step 3; since an error is found here, also provide
detailed critique and correction guideline)...
</analysis_3>
<conclusion>
Incorrect
</conclusion>
```

- *Note that the analyses of steps 4 and 5 is skipped as step 3 has been found to contain an error.*

---

*The following is the abstract visual reasoning problem and its corresponding solution, for you to review and verify:*
*[Abstract Visual Reasoning Problem]*

```
<abstract_visual_reasoning_problem>
{{ABSTRACT_VISUAL_REASONING_PROBLEM}}
</abstract_visual_reasoning_problem>
```

*[Solution]*

```
<solution>
{{SOLUTION}}
</solution>
```

*Remember:*

- *The <conclusion> tag must contain either Correct or Incorrect, with no additional text or punctuation.*

---

GPT-4.1 Prompt for Rephrasing template perception in PuzzleVQA for PerceptionBench

*You are an expert assistant for creating training data for vision-language models.*
*Your task is to take an image, a question about the image, and its template based caption that provides visual perception of the image relevant to answering the question.*
*You must generate rephrased captions that are fluent, natural and contain all the information from the template based caption.*

*Your Task is to not to answer the given question but to rephrase captions so they are more natural than the template based captions.*

*Guidelines:*

1. *Maintain fluency: Captions must remain human-like and grammatically correct.*
2. *Do not change the content: Rephrased captions must remain correct and contain the information from the original caption.*
3. *Generate only the Rephrased Caption*

---

**GPT-4.1 Prompt for Mutating Perception in PuzzleVQA for PerceptionBench**

*You are an expert assistant for creating training data for vision-language models.*
*Your task is to take an image, a question about the image, and its correct caption that answers the question.*
*You must generate mutated captions that are fluent and natural but contain subtle factual errors with respect to the image and question context.*
*Your Task is to not to answer the given question but to generate mutated captions that can serve as negative samples.*

*Guidelines:*

1. *Maintain fluency: Captions must remain human-like and grammatically correct.*
2. *Introduce only subtle mistakes relevant to the question:*
   - *Change an object to a visually similar but incorrect one (dog → cat, apple → orange).*
   - *Alter attributes such as color, size, number, position, or action.*
   - *Slightly modify spatial relations or other details that are relevant to answering the question.*
3. *Minimal edits: Only introduce one subtle error per mutated caption.*
4. *Do not generate nonsense: Captions must remain realistic and plausible, even if incorrect.*
5. *Contrastive purpose: Mutated captions are intended as negative samples that could mislead a model answering the same question. Always ensure the error is contextually relevant to the question.*

---

**Gemma Policy Prompt for TTS Generation**

*[BOS]<start_of_turn>user*
*You are an expert visual analyst who solves complex visual reasoning problems by systematically connecting what you observe in images to the specific requirements of each problem. First, identify and inventory all problem-relevant visual elements through a step-by-step perceptual process, documenting their spatial relationships, object identities, and perceptual details with precision at each stage of observation. Then, construct a step-by-step logical pathway that explicitly shows how each visual observation leads to your solution, explaining why each detail matters and citing specific image evidence at every step. Your analysis will be evaluated by a teacher on two criteria: \*\*Visual Elements\* (accuracy of perception, spatial understanding, and object recognition) and \*Reasoning\*\* (logical consistency, valid deductive steps, and sound analytical conclusions). Multiple solution pathways will be explored in parallel at each step, building upon previous reasoning to generate diverse approaches, with the teacher scoring each path to accept or reject based on visual accuracy and logical validity. This approach increases the odds of reaching a correct final answer by exploring and focusing on perception and reasoning traces that are valid and correct while ignoring incorrect ones. This makes your precise step-by-step perception and reasoning essential for identifying the most promising solution.*

*<start_of_image> Question: {{Question}} Options: {{Options}}*
*If you are uncertain or the problem is too complex, make a reasoned guess based on the information provided. Avoid repeating steps indefinitely—provide your best guess even if unsure. Determine whether to think step by step based on the difficulty of the question, considering all relevant information before answering. If you do need to think step by step, first, carefully observe and describe everything you see: the image itself and how it connects to the problem being presented. Then, work through your reasoning step by step to arrive at your answer. *Your teacher will review your descriptions of visual elements to ensure you're observing all relevant details accurately, and will critique each step of your reasoning to provide guidance and ensure you're on the right track.* Put your final answer within \boxed{}. If multiple-choice options for answers to the question are provided, select the alphabetical letter of the corresponding best option within \boxed{} when you are ready to provide your final answer. <end_of_turn>*
*<start_of_turn>model*

---

**Qwen Policy Prompt for MathVision**

*< |im_start| >system*
*You are a helpful assistant.< |im_end| >*
*< |im_start| >user*
*< |vision_start| >< |image_pad| >< |vision_end| > Please solve the problem step by step and put your answer in one \boxed{}". If it is a multiple choice question, only one letter is allowed in the \boxed{}".*
**{{Question}}**
*< |im_end| >*
*< |im_start| >assistant*

---

**Gemma Policy Prompt for MathVision**

*[BOS]<start_of_turn>user*
*You are a helpful assistant.*
*<start_of_image> Please solve the problem step by step and put your answer in one "\boxed{}". If it is a multiple choice question, only one letter is allowed in the "\boxed{}".*
**{{Question}}**
*<end_of_turn>*
*<start_of_turn>model*

---

**QWEN-VL-PRM-3B and QWEN-VL-PRM-7B Prompt for Step Level Scoring**

***You are a process supervision model for visual reasoning tasks. You will receive an image and an image-based problem statement, followed by solution steps to evaluate.***
*First round: problem statement and first solution step.*
*Subsequent rounds: one new step per round.*
*Assess the cumulative correctness of the entire solution up to each step.*
*Evaluation Criteria:*

1. **Visual Accuracy**: *Are visual elements from the image correctly identified (shapes, colors, positions, quantities, spatial relationships)?*

2. **Logical Validity**: *Do all inferences and calculations follow correctly from the image and previous steps?*

*Response:*

- *"+" if correct up to this step*

- *"-" if any error exists up to this step*

*Only respond with "+" or "-". No explanations.*
*An error in any step invalidates all subsequent steps.*

## A.5 Training Details

**Details** In this Section, we will provide details on training. We utilize the HuggingFace TRL library with PyTorch torchrun for distributed training to perform full parameter fine-tuning on Qwen2.5-VL-3B/7B-Instruct to create QWEN-VL-PRM-3B and QWEN-VL-PRM-7B.

| Parameter | Value |
|---|---|
| Base Model | Qwen2.5-VL-3B/7B-Instruct |
| Torch Data type | bfloat16 |
| Attention Implementation | flash attention 2 |
| Per-device Train Batch Size | 2 |
| Gradient Accumulation Steps | 4 |
| Learning Rate | 1.0e-05 |
| Weight Decay | 1.0e-04 |
| Number of Training Epochs | 2 |
| LR Scheduler Type | cosine |
| Max Gradient Norm | 1.0 |
| Warmup Ratio | 0.05 |
| Seed | 100 |
| BF16 | true |
| Optimizer | adam |
| Adam Beta 1 | 0.9 |
| Adam Beta 2 | 0.95 |
| Gradient Checkpointing | True |
| Tune Vision Encoder | False |

Table 12: Training Configuration for QWEN-VL-PRM-3B and QWEN-VL-PRM-7B.

**Effect of resizing images on performance before training** Similar to dynamic resizing of images using image processors at inference time recommended by Qwen-2.5-VL for optimized inference performance, we experiment with applying the same dynamic image resizing procedure to all images in our training dataset and observed performance improvements on all evaluation benchmarks.

**Effect of tuning vision layers on performance** We experimented with full parameter fine-tuning, including and excluding the layers of the vision encoder and concluded that excluding layers relating to the vision encoder consistently results in better performance for QWEN-VL-PRM-3B and QWEN-VL-PRM-7B on all benchmarks. Hence, we tune all parameters when training and freeze all vision layers.

## A.6 Test-time Scaling Decoding Details

**Details** In this Section, we will provide details used for test-time scaling decoding when running the following policy models with QWEN-VL-PRM-3B and QWEN-VL-PRM-7B on all evaluation benchmarks. We use vLLM version 0.10.1.1 with V1 engine, transformers version 4.55.2 and flash-attn version 2.8.0.post2 for inference. We experience varying results across models when different versions of these key packages are used, and found fixing these versions resulted in the best overall performance.

| Parameter | Value |
|---|---|
| N | 1 |
| Top P | 0.001 |
| Top K | 1 |
| Temperature | 0.01 |
| Max New Tokens | 16384 |
| Repetition Penalty | 1.0 |

Table 15: Evaluation decoding configuration for baseline model evaluation.

| Parameter | Value |
|---|---|
| Base Model | Qwen/Qwen2.5-VL-3B/7B/32B-Instruct |
| N | 16 |
| Top P | 0.8 |
| Top K | 20 |
| Temperature | 0.7 |
| Max New Tokens | 16384 (One-shot), 8196 (Greedy) |
| Repetition Penalty | 1.05 |

Table 13: Evaluation decoding configuration for Qwen family policy models.

| Parameter | Value |
|---|---|
| Base Model | google/gemma-3-12B/27B-it |
| N | 16 |
| Top P | 0.95 |
| Top K | 64 |
| Temperature | 0.7 |
| Max New Tokens | 16384 (One-shot), 8196 (Greedy) |
| Repetition Penalty | 1.0 |

Table 14: Evaluation decoding configuration for Gemma family policy models.

### A.7 BASELINE MODEL EVALUATION DECODING DETAILS

**Details** In this Section, we will provide details used for baseline model evaluation when running the following policy models without TTS on all evaluation benchmarks.

### A.8 EVALUATION DATA EXAMPLES

In this section, we provide some examples from the TTS evaluation with correct and incorrect candidate solutions along with their PRM scores.

### A.8.1 ALGOPUZZLEVQA

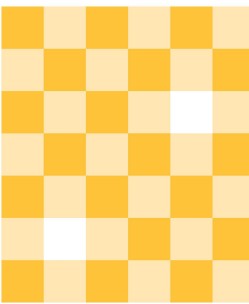

**Question:** The checkerboard shown in the image was originally of 7 * 6 in dimension having a total of 42 squares. It uses two colours of squares, one light yellow and one dark yellow, in a chequered pattern. Two of the squares have been removed from the board in the position of the white coloured cells, as shown in the image. You have 20 dominoes of size 2 * 1. You can use them as is or you can rotate them to use as a 1 * 2 domino. Is it possible to place all the 20 dominoes in the checkerboard to exactly cover all the remaining 40 squares?

**Incorrect Candidate Solution:**
### Step 1: Analyze the Checkerboard Pattern
The checkerboard in the image has a standard alternating pattern of light yellow and dark yellow squares. The board is missing two squares, which are both white (light yellow) in color. The original dimensions of the board were $7 \times 6$, meaning it had 42 squares. After removing two squares, there are 40 squares left.

### Step 2: Count the Number of Light Yellow and Dark Yellow Squares
In a standard checkerboard pattern:
- The number of light yellow and dark yellow squares alternates in a consistent manner.
- For a $7 \times 6$ board, the total number of squares is 42. Since the pattern alternates, there will be an unequal distribution of light yellow and dark yellow squares because the total number of squares is even, but the dimensions are odd $\times$ even.
- Specifically, in a $7 \times 6$ board:
- There will be 22 light yellow squares.
- There will be 20 dark yellow squares.

When two light yellow squares are removed:
- The number of light yellow squares becomes $22 - 2 = 20$.
- The number of dark yellow squares remains 20.

Thus, after removing the two light yellow squares, the board has:
- 20 light yellow squares.
- 20 dark yellow squares.

### Step 3: Understand the Domino Placement
Each domino covers exactly 2 squares. Since we have 20 dominoes, they will cover:

$$20 \times 2 = 40 \text{ squares}$$

This matches the number of remaining squares on the board.

### Step 4: Check the Color Distribution
Each domino must cover one light yellow square and one dark yellow square because the domino is $2 \times 1$ or $1 \times 2$ and the checkerboard alternates colors. For the dominoes to fit perfectly:
- The number of light yellow squares must equal the number of dark yellow squares.

From Step 2, we know:
- There are 20 light yellow squares.
- There are 20 dark yellow squares.

Since the number of light yellow squares equals the number of dark yellow squares, it is possible to place the dominoes such that each domino covers one light yellow square and one dark yellow square.

### Step 5: Conclusion
Given that the number of light yellow squares equals the number of dark yellow squares, and each domino covers one light yellow square and one dark yellow square, it is possible to place all 20 dominoes to cover the remaining 40 squares.

### Final Answer:\boxed{A}
**PRM Score: 0.85**

**Correct Candidate Answer:**
### Step 1: Analyze the Checkerboard Pattern
The checkerboard initially has dimensions of $7 \times 6$, which totals 42 squares. The squares alternate between two colors: light yellow and dark yellow, forming a standard checkerboard pattern. In such a pattern:
- Half of the squares are light yellow.
- Half of the squares are dark yellow.

Since there are 42 squares in total:
- There are 21 light yellow squares.
- There are 21 dark yellow squares.

### Step 2: Identify Removed Squares
Two squares have been removed from the checkerboard, and these squares are marked as white in the image. By examining the image:
- Both removed squares are light yellow.

This means:
- After removing the two light yellow squares, there are now 19 light yellow squares.
- The number of dark yellow squares remains unchanged at 21.

### Step 3: Understand the Domino Placement
A domino covers exactly 2 adjacent squares. Importantly:
- Each domino must cover one light yellow square and one dark yellow square because the checkerboard alternates colors.

### Step 4: Check the Feasibility of Covering the Board with Dominoes
To cover the entire board with 20 dominoes:
- Each domino covers 2 squares, so 20 dominoes will cover $20 \times 2 = 40$ squares.
- For the dominoes to fit perfectly, the number of light yellow squares must equal the number of dark yellow squares.

However, after removing the two light yellow squares:
- Light yellow squares: 19
- Dark yellow squares: 21

This creates an imbalance:
- There are 2 more dark yellow squares than light yellow squares.

Since each domino must cover one light yellow square and one dark yellow square, it is impossible to pair all the dark yellow squares with light yellow squares due to this imbalance.

### Step 5: Conclusion
Given the imbalance in the number of light yellow and dark yellow squares, it is not possible to place all 20 dominoes to cover the remaining 40 squares.

### Final Answer:\boxed{B}
**PRM Score: 0.97**

A.8.2 PUZZLEVQA

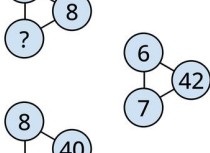

**Question: What is the missing number of the part denoted with a question mark?**

**Incorrect Candidate Solution:** ### Step-by-Step Analysis

### **Step 1: Observing the Image**
The image consists of two sets of interconnected circles, each containing numbers. The connections between the circles suggest a relationship or pattern that needs to be identified.

#### **Left Set:**
- Circles contain the numbers: 4, 8, ?, 8, 5, 40.
- The question mark indicates a missing number that we need to determine.

#### **Right Set:**
- Circles contain the numbers: 6, 7, 42.
- This set appears complete and may help us identify the pattern.

### **Step 2: Identifying Patterns**
To solve the problem, we need to determine the relationship between the numbers in each set. Let's analyze both sets step by step.

#### **Analyzing the Right Set (Complete):**
- Numbers: 6, 7, 42.
- Relationship: It appears that the product of the first two numbers equals the third number.
- $6 \times 7 = 42$.

This suggests a multiplication pattern.

#### **Applying the Pattern to the Left Set:**
- Numbers: 4, 8, ?, 8, 5, 40.
- We need to verify if a similar multiplication pattern holds.

#### **Testing the Multiplication Pattern:**
1. **Top Pair (4 and 8):**
- $4 \times 8 = 32$.
- However, there is no 32 in the image, so this might not be the direct pattern.

2. **Middle Pair (? and 8):**
- Let the missing number be $x$.
- According to the pattern, $x \times 8$ should equal the next number in the sequence.

3. **Bottom Pair (8 and 5):**
- $8 \times 5 = 40$.
- This matches the given number 40, confirming the multiplication pattern.

### **Step 3: Solving for the Missing Number**
From the bottom pair, we confirmed the multiplication pattern. Now, let's apply it to the middle pair:

- The middle pair is ? and 8.
- The next number after 8 is 40 (from the bottom pair).
- Therefore, $? \times 8 = 40$.

Solving for ?:

$$? = \frac{40}{8} = 5$$

### **Step 4: Verifying the Solution**
- If the missing number is 5, then the sequence becomes: 4, 8, 5, 8, 5, 40.

- Checking the multiplication:
- $8 \times 5 = 40$ (correct).
- The pattern holds.

### **Final Answer:**
The missing number is 5.\boxed{c}
**PRM Score: 0.85**

**Correct Candidate Solution:** ## Step 1: Analyze the Image and Identify Patterns

The image consists of three sets of numbers connected by lines. Each set appears to follow a specific mathematical relationship. Let's break down the sets:

### Set 1:
- Numbers: 4, 8, ?
- Connection: The question mark indicates a missing value we need to determine.

### Set 2:
- Numbers: 6, 7, 42
- Connection: These numbers are connected, suggesting a relationship between them.

### Set 3:
- Numbers: 8, 5, 40
- Connection: These numbers are also connected, suggesting another relationship.

## Step 2: Examine the Relationships in Sets 2 and 3

### Set 2: 6, 7, 42
- Observe the relationship between 6, 7, and 42.
- A common operation that could relate these numbers is multiplication.
- Check: $6 \times 7 = 42$.
- This confirms that the relationship in Set 2 is multiplication.

### Set 3: 8, 5, 40
- Observe the relationship between 8, 5, and 40.
- Again, multiplication seems plausible.
- Check: $8 \times 5 = 40$.
- This confirms that the relationship in Set 3 is also multiplication.

## Step 3: Apply the Observed Pattern to Set 1

### Set 1: 4, 8, ?
- Based on the patterns observed in Sets 2 and 3, the relationship appears to be multiplication.
- Assume the missing number (?) is related to 4 and 8 through multiplication.
- Check: $4 \times ? = 8$.

## Step 4: Solve for the Missing Number

- From the equation $4 \times ? = 8$:

$$? = \frac{8}{4} = 2$$

## Step 5: Verify the Solution

- The pattern of multiplication holds across all sets:

- Set 2: $6 \times 7 = 42$
- Set 3: $8 \times 5 = 40$
- Set 1: $4 \times 2 = 8$

## Final Answer:
\boxed{B}
**PRM Score: 0.98**

A.8.3 MMMU DEV SET

|  | Average Cost per Unit |
|---|---|
| Direct materials | $10 |
| Direct labor | 9 |
| Indirect materials | 3 |
| Fixed manufacturing overhead | 6 |
| Variable manufacturing overhead | 2 |
| Fixed selling and administrative expenses | 8 |
| Variable sales commissions | 14 |

**Question:** Baxter Company has a relevant range of production between 15,000 and 30,000 units. The following cost data represents average variable costs per unit for 25,000 units of production. If 30,000 units are produced, what are the per unit manufacturing overhead costs incurred? (A). $6 (B). $7 (C). $8 (D). $9

**Incorrect Candidate Solution:**
### Step 1: Analyze the Problem and Identify Key Information
The problem asks us to determine the **per unit manufacturing overhead costs** if 30,000 units are produced. Manufacturing overhead includes both **fixed manufacturing overhead** and **variable manufacturing overhead**.

From the table:
- **Fixed manufacturing overhead**: $6 per unit
- **Variable manufacturing overhead**: $2 per unit

The total manufacturing overhead per unit is the sum of these two components:
Total manufacturing overhead per unit = Fixed manufacturing overhead per unit + Variable manufacturing overhead per unit

### Step 2: Understand the Concept of Fixed and Variable Costs
- **Fixed manufacturing overhead** remains constant per unit regardless of the number of units produced within the relevant range (15,000 to 30,000 units).
- **Variable manufacturing overhead** varies directly with the number of units produced.

Since the fixed manufacturing overhead is given as $6 per unit and the variable manufacturing overhead is given as $2 per unit, these values should remain consistent within the relevant range.

### Step 3: Calculate Total Manufacturing Overhead per Unit
Using the formula:
Total manufacturing overhead per unit    =    Fixed manufacturing overhead per unit    + Variable manufacturing overhead per unit

Substitute the given values:
Total manufacturing overhead per unit = 6 + 2 = 8

### Step 4: Verify the Relevant Range
The problem specifies that the relevant range of production is between 15,000 and 30,000 units. Since 30,000 units fall within this range, the given cost data (fixed and variable overhead rates) can be applied directly without any adjustments.

### Step 5: Match the Answer to the Options
The total manufacturing overhead per unit is $8. Comparing this to the provided options:
- A. $6
- B. $7
- C. $8
- D. $9

The correct answer is
\boxed{C}.

### Final Answer:
\boxed{C}
**PRM Score: 0.86**

**Correct Candidate Solution:**
### Step 1: Analyze the Problem and Identify Key Information
The problem asks us to determine the **per unit manufacturing overhead costs** for 30,000 units of production. Manufacturing overhead consists of both **fixed manufacturing overhead** and **variable manufacturing overhead**. We are given the following information:

- **Fixed manufacturing overhead**: $6 per unit for 25,000 units.
- **Variable manufacturing overhead**: $2 per unit for 25,000 units.
- Relevant range of production: 15,000 to 30,000 units.

We need to calculate the per unit manufacturing overhead for 30,000 units, considering the behavior of fixed and variable costs.

—

Step 2: Understand Cost Behavior
1. **Fixed Manufacturing Overhead**:
- Fixed costs remain constant in total but vary inversely with the number of units produced. For example, if the total fixed manufacturing overhead is constant, the per-unit fixed overhead decreases as production increases.
- Given: $6 per unit for 25,000 units.
- Total fixed manufacturing overhead = $6 × 25,000 = $150,000.
- For 30,000 units, the per-unit fixed overhead = $150,000 ÷ 30,000 = $5.

2. **Variable Manufacturing Overhead**:
- Variable costs remain constant per unit regardless of the level of production.
- Given: $2 per unit for 25,000 units.
- For 30,000 units, the per-unit variable overhead remains $2.

—

Step 3: Calculate Total Per Unit Manufacturing Overhead for 30,000 Units

Manufacturing overhead = Fixed manufacturing overhead + Variable manufacturing overhead.
- Per unit fixed manufacturing overhead for 30,000 units = $5.
- Per unit variable manufacturing overhead for 30,000 units = $2.
- Total per unit manufacturing overhead = $5 + $2 = $7.

—

Step 4: Verify the Answer
- The fixed manufacturing overhead decreases from $6 per unit (at 25,000 units) to $5 per unit (at 30,000 units) because the total fixed cost is spread over more units.
- The variable manufacturing overhead remains constant at $2 per unit.
- Adding these together gives the total per unit manufacturing overhead of $7.

—

Final Answer:\boxed{B}
**PRM Score: 0.94**

### A.8.4 MATHVISTA TESTMINI

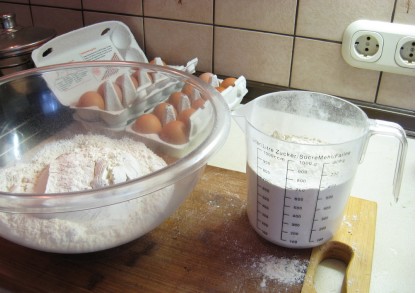

**Question:** what is the total volume of the measuring cup?

**Incorrect Candidate Solution:**
Step 1: Observing the Image
- **Measuring Cup**: The image shows a transparent measuring cup filled with flour. The cup has markings indicating measurements in both milliliters (ccm) and grams (g).
- **Markings on the Measuring Cup**:
- The markings are clearly visible and include both metric units (e.g., 1000 ccm, 900 ccm, etc.) and gram equivalents (e.g., 800 g, 700 g, etc.).
- The topmost marking visible is ¨1000 ccm,¨which corresponds to ¨800 g.¨
- **Flour Level**: The flour in the measuring cup reaches the ¨1000 ccm¨mark.

Step 2: Connecting Visual Elements to the Problem
- The question asks for the **total volume of the measuring cup in grams**.
- From the markings on the measuring cup, we can see that the flour level aligns with the ¨1000 ccm¨mark, which corresponds to ¨800 g.¨

Step 3: Reasoning Through the Problem
- The measuring cup is designed to measure both volume (in ccm) and weight (in grams) for ingredients like flour.
- The markings indicate that 1000 ccm of flour weighs 800 g. Since the flour level in the cup reaches the 1000 ccm mark, the total volume of the measuring cup in grams is directly given by the corresponding gram measurement.

Final Answer:
\boxed{800}
**PRM Score: 0.86**

**Correct Candidate Solution:**
Step 1: Observing the Image
- **Measuring Cup**: The measuring cup is clearly visible on the right side of the image. It has markings indicating measurements in grams and other units.
- **Markings on the Measuring Cup**: The markings show a scale from 0 to 1000 grams. The largest marking corresponds to 1000 grams.
- **Context**: The measuring cup is likely used for baking, as flour and eggs are also present, suggesting preparation for a recipe.

Step 2: Analyzing the Measuring Cup
- The measuring cup is designed for measuring dry ingredients, such as flour, sugar, or powdered substances.
- The largest marking on the cup is labeled Ï000 g,ẅhich indicates the maximum capacity of the measuring cup.

Step 3: Reasoning
- The question asks for the total volume of the measuring cup in grams.
- From the markings on the cup, the maximum capacity is explicitly labeled as 1000 grams.
- There is no ambiguity in the markings, and the question does not require any additional calculations or estimations.

Final Answer:
\boxed{1000}
**PRM Score: 0.99**

A.8.5   MATHVISION TEST

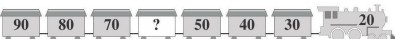

**Question:** Which number should be written in place of the question mark?

**Incorrect Candidate Solution:**
Okay, let's analyze this visual reasoning problem.

**1. Visual Elements Inventory:**

• **Overall Structure:** The image depicts a train composed of connected boxcars and a locomotive.
• **Boxcars:** There are 9 boxcars in total. Each boxcar is a rectangular prism shape.
• **Numbers:** Each boxcar has a number printed on its side. The numbers are: 90, 80, 70, ?, 50, 40, 30, 20.
• **Locomotive:** The locomotive is at the right end of the train. It has the number 20 on it.
• **Spatial Arrangement:** The boxcars are arranged in a linear sequence, connected to each other. The locomotive is connected to the boxcar immediately to its left. The question mark is on the fourth boxcar from the left.

**2. Reasoning Pathway:**

The numbers on the boxcars seem to follow a decreasing pattern. Let's analyze the sequence:

• 90, 80, 70, ?, 50, 40, 30, 20
• The difference between 90 and 80 is 10.
• The difference between 80 and 70 is 10.
• Following this pattern, the number before 50 should be 50 - 10 = 40.

Therefore, the missing number should be 40.

**3. Verification and Alternative Pathways:**

Let's see if there's an alternative pattern. Considering the locomotive's number (20) and the overall decreasing trend, it's reasonable to assume the numbers are decreasing by 10.

• $90 \rightarrow 80 \rightarrow 70 \rightarrow 60 \rightarrow 50 \rightarrow 40 \rightarrow 30 \rightarrow 20$.

This sequence confirms that 60 is the missing number.

**4. Final Answer:**

Given the established pattern of decreasing numbers by 10, the missing number should be 60.

\boxed{60}
**PRM Score: 0.26**

---

**Correct Candidate Solution:**
Okay, let's analyze this visual reasoning problem.

**Visual Elements:**

1. **Image Description:** The image shows a train composed of several connected train cars. Each train car has a number written on its side.
2. **Numbers:** The numbers visible on the train cars are: 90, 80, 70, ?, 50, 40, 30, and 20. The locomotive (engine) has the number 20.
3. **Spatial Arrangement:** The numbers are arranged sequentially from left to right along the train cars. There is a question mark on one of the cars, indicating a missing number.
4. **Pattern Observation:** The numbers seem to be decreasing. 90, 80, 70...
5. **Locomotive:** The locomotive, at the right end, has the number 20.

**Reasoning:**

Let's explore a possible pattern.

• **Pathway 1: Arithmetic Sequence:** The sequence appears to be decreasing by 10.
• 90 - 10 = 80
• 80 - 10 = 70
• Following this pattern, the next number should be 70 - 10 = 60.
• 60 - 10 = 50
• 50 - 10 = 40
• 40 - 10 = 30
• 30 - 10 = 20 (Locomotive)

This arithmetic sequence looks consistent with the provided numbers.

**Final Answer:**

Based on the observed pattern, the missing number should be 60.

\boxed{60}
**PRM Score: 0.98**

### A.9 ANALYSIS OF ERRORS BETWEEN ONE SHOT AND STEP AGGREGATION SCORING

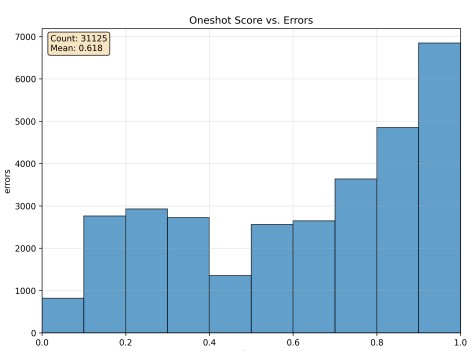 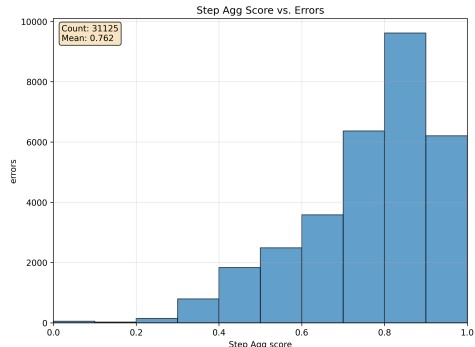

Figure 6: Comparison of error distribution histograms across two scoring methods (One shot score on the left and Step Aggregate Scor on the right), showing the frequency of incorrect samples per score bin.

To better understand the performance differences between the two scoring methods, we analyzed the distribution of errors with respect to their assigned scores. Figure 6 shows histograms of incorrect samples for the Oneshot and Step Aggregate scoring approaches.

The results reveal two key trends. First, the Oneshot method tends to assign lower scores to incorrect samples on average (mean score 0.62), with a relatively even spread of errors across the score bins. This indicates that when the model produces an error, the assigned confidence is often more moderate, reflecting some degree of uncertainty.

In contrast, the Step Aggregate method produces a markedly different error profile. Incorrect samples under this scoring scheme are skewed towards the higher end of the score range (mean score 0.76), with a large concentration of errors receiving scores close to 1.0. This indicates that the Step Aggregate approach is more prone to overestimating its confidence when the output is incorrect. A likely reason is that even when a final solution is wrong, it may consist largely of correct intermediate steps, with only a few erroneous steps leading to the incorrect answer. As a result, the aggregation process inflates the overall score, causing erroneous samples to receive higher scores on average.

This phenomenon may explain why the Oneshot scoring approach is more effective than Step Aggregation for selecting the best candidate solution. Because Step Aggregation assigns disproportionately high scores to incorrect solutions, driven by the sparsity of erroneous steps within largely correct reasoning chains, it risks favoring poor solutions that nonetheless appear strong under the aggregated metric. In contrast, Oneshot scoring evaluates the solution holistically, resulting in a more reliable selection from the candidate solutions.

## B LLMs USAGE

LLMs were used to polish the writing.

