# OpenReview forum: "Training Vision-Language Process Reward Models for Test-Time Scaling in Multimodal Reasoning: Key Insights and Lessons Learned"
_ICLR.cc/2026/Conference — ICLR 2026 Conference Withdrawn Submission_

### Official Review · Reviewer_G3VA · 2025-10-30

**Soundness:** 1
**Presentation:** 2
**Contribution:** 1
**Rating:** 2
**Confidence:** 4

**Summary:**

This work addresses the limitations of existing Vision-Language Process Reward Models (VL-PRMs), which rely on noisy MCTS-based data and lack effective visual grounding supervision. The authors propose a hybrid data synthesis method combining MCTS with strong VLM judgments, introduce perception-focused supervision to detect visual grounding errors, and explore test-time scaling strategies. Experiments on five multimodal benchmarks (MMMU, PuzzleVQA, AlgoPuzzleVQA, MathVista, MathVision) show that the proposed VL-PRM improves reasoning reliability, reveals latent abilities in VLMs, and achieves strong performance without task-specific training.

**Strengths:**

The paper is clearly written and well-structured, with extensive experiments provided to demonstrate the effectiveness of the proposed method.

**Weaknesses:**

On Methodology:
1. The core components of the proposed method have been previously explored in prior work. Techniques for data construction, as well as approaches for evaluating reasoning processes—such as using process reward models (PRMs) to assess intermediate steps, generating fine-grained reasoning trajectories, or leveraging bounding box detection for spatial accuracy—have already been extensively studied in the literature. The current work does not sufficiently differentiate its methodological contributions from these existing approaches.

On Experiments:
2. If positioned as an analytical study, the paper lacks novel and insightful findings. The conclusions drawn from the experiments are largely consistent with established understanding and do not provide significant new discoveries or deeper mechanistic insights into multimodal reasoning or model merging.

[1] MM-PRM:Enhancing Multimodal Mathematical Reasoning with Scalable Step-Level Supervision, 2025.06
[2] Let’s Verify Step by Step, 2023.05
[3] R-PRM: Reasoning-Driven Process Reward Modeling, 2025.03
[4] Semi-off-Policy Reinforcement Learning for Vision-Language Slow-thinking Reasoning. 2025.7.22

**Questions:**

See Weakness

---

### Official Review · Reviewer_o8TV · 2025-11-01

**Soundness:** 2
**Presentation:** 1
**Contribution:** 3
**Rating:** 4
**Confidence:** 4

**Summary:**

The paper explores the design space of VL-PRMs, proposing a hybrid data construction (MCTS + VLM judge), adding perception-focused supervision, and comparing several test-time scaling (TTS) strategies (guided greedy, one-shot, step-score aggregation). Experiments across five multimodal benchmarks and VisualProcessBench show non-trivial gains and several diagnostic insights.

**Strengths:**

Hybrid supervision & cleaner labels: Using a strong VLM judge on top of MCTS to label step correctness is well-motivated and empirically investigated; this is a sensible response to noisy MC-score labeling.

Perception-focused process supervision: Explicitly separating perception from higher-level reasoning and training VL-PRMs to catch visual grounding errors is novel and useful.

**Weaknesses:**

Prompt dependence. The policy side consistently uses structured reasoning prompts (perception first, then reasoning). In practice, we sometimes need alternative prompting styles; if the method only works with your structural template, applicability feels constrained.

Figures readability issues: Many tables/figures are hard to read due to very small fonts. Please enlarge fonts and line weights.

Marginal improvement: Improvements are marginal and do not clearly surpass a majority voting. On MMMU, MathVista benchmarks, the improvements are also marginal.

**Questions:**

“Akin to ORM”: You state the method is akin to ORM (one-shot selection). Why is the simplest strategy the strongest? If we only need a single final selection, what is the meaning of a step-level PRM that evaluates every intermediate step?

Across-benchmark variance: Your PRM yields more than 10% gains on some benchmarks but less than 2% on some others. Could you analyze the causes? Also, can you evaluate on the challenging setting—e.g. Humanity Last Exam benchmark—with GPT-5-mini under a Best-of-N (BoN) protocol to test whether the gains persist for strong models?

BoN details and scaling: How many distinct responses are sampled for your BoN results? As N increases, how does PRM-guided selection scale relative to majority voting (MV@N)?

---

### Official Review · Reviewer_ookY · 2025-11-01

**Soundness:** 3
**Presentation:** 3
**Contribution:** 2
**Rating:** 4
**Confidence:** 3

**Summary:**

This paper studies vision‑language process reward models (VL‑PRMs) and how to best use them at inference time. The authors (1) build VL‑PRM300K, a process‑supervision dataset created with MCTS rollouts plus an external judge (o4‑mini) to label each step; the traces are structured into explicit perception steps followed by reasoning steps; (2) fine‑tune 3B and 7B Qwen‑based PRMs on this data; and (3) compare several test‑time scaling (TTS) strategies—guided greedy, step‑score aggregation, and one‑shot (whole‑solution) scoring. The paper reports that one‑shot/ORM‑like usage of a PRM generally works best; perception supervision is critical; and smaller PRMs can rival larger ones for error detection. Evaluations span MMMU, PuzzleVQA, AlgoPuzzleVQA, MathVista, MathVision, plus VisualProcessBench for step‑level error detection. Gains are largest on abstract/algorithmic datasets (PuzzleVQA and AlgoPuzzleVQA across backbones), modest on MMMU and 3–5%.

**Strengths:**

1. Clear problem framing and expanded design space. The paper explicitly contrasts MCTS‑only labeling (e.g., VisualPRM) with hybrid MCTS+LLM‑judge labeling and adds perception‑step supervision—two axes that are under‑explored for VL‑PRMs.

2. Study of TTS strategies. The comparison of guided greedy vs. step aggregation vs. one‑shot is systematic and revealing: one‑shot consistently wins on reasoning‑heavy tasks (by 4–9% over greedy on PuzzleVQA/AlgoPuzzleVQA; Fig. 2; Table 6). This is a practical take‑home result for practitioners.

3. Cross‑backbone generality. The PRMs trained on Qwen are used to score solutions from Qwen 7B/32B and Gemma3‑12B/27B, with consistent gains. This strengthens claims of policy‑agnostic utility.

4. Small PRMs can be strong validators. On VisualProcessBench, the 3B PRM achieves ~62 macro‑F1, outperforming its 7B counterpart on several subsets and comparing favorably to GPT‑4o‑Mini. This is both cost‑relevant and interesting scientifically.

**Weaknesses:**

1. The novelty is incremental: (i) hybrid MCTS+LLM‑judge labeling; (ii) explicit perception‑step supervision; (iii) a comprehensive TTS comparison where one‑shot (PRM as ORM) emerges as best. These are meaningful engineering insights, but conceptually less radical than introducing a learning objective.

2. The claim that VL‑PRM300K is the first multimodal process dataset with predominantly abstract reasoning (Appx. A.1) is plausible but narrow; though earlier PRM datasets exist (VisualPRM400K) is math‑heavy.

3. Two datasets are custom subsets (first 50 puzzles per type for PuzzleVQA and AlgoPuzzleVQA), which may introduce selection bias and makes comparison to prior art harder. Stronger evidence would utilise official test splits / held‑out hidden servers where available, or practice from other works for pair apple-to-apple comparison.

4. Inconsistency about post‑error steps. In Line 169 the authors say that after a negative step they “discard all subsequent steps”; later, Line 377 says the data is structured so that all subsequent steps are “marked as negative” after the first error. Please clarify whether later steps are dropped or labeled negative.

5. Risk of label/model bias. Using o4‑mini as the judge to train the PRM and then evaluating against outcomes that may correlate with that judge might bias labels.

**Questions:**

The paper is titled as an empirical-like work, and indeed, practical solutions and recipes are introduced. However, I'm still unsure/confused about the novelty, experiments, and claims, as listed in the weakness section. Kindly clarify the points there in the rebuttal.

---

### Official Review · Reviewer_X3V3 · 2025-11-04

**Soundness:** 2
**Presentation:** 2
**Contribution:** 3
**Rating:** 6
**Confidence:** 3

**Summary:**

This paper extends Process Reward Models (PRMs) to the multimodal setting by developing Vision-Language PRMs (VL-PRMs) that provide step-level feedback to improve reasoning reliability in Vision-Language Models (VLMs). The authors propose three main innovations: (1) data synthesis framework combining Monte Carlo Tree Search (MCTS) with judgments from a strong VLM to reduce label noise; (2) perception-focused supervision to explicitly detect visual grounding errors; and (3) a systematic study of test-time scaling strategies guided by PRMs. Experiments across five multimodal reasoning benchmarks reveal valuable insights into how perception-level supervision helps.

**Strengths:**

1. Perception reasoning plays a crucial role, and this insight is effectively leveraged in the dataset design and PRM training, leading to substantial performance gains.

2. This is a crucial finding, demonstrating that VL-PRMs used as Outcome Reward Models (ORMs) during test-time scaling (TTS) can outperform VL-PRM-guided process step selection. This highlights the advantage of leveraging finer-grained reward at the end for improved reasoning performance.

3. This is an insightful finding, the authors hypothesize that the inclusion of a substantial number of abstract reasoning and general VQA tasks enhances the model’s overall logical reasoning ability, thereby improving test-time scaling (TTS) performance across diverse domains.

**Weaknesses:**

1) "In particular, abstract reasoning tasks highlight human strengths but expose persistent weaknesses in VLMs" – Please add references for this claim.

2) Footnote 1 appears to be missing or incomplete.

3) It would strengthen the paper to include a brief manual analysis or discussion of cases where the judge model (GPT-4.1) fails to assign correct labels.

4) The paper should elaborate on how the dataset quality and label accuracy are ensured. Were any manual inspections or spot-checks performed to validate the automatically generated annotations? If so, providing statistics or qualitative examples would improve transparency.

5) Please define what is meant by abstract reasoning and perception-based reasoning when these terms are first introduced. Providing concise definitions or examples would help readers clearly distinguish between the two reasoning types and understand their respective roles in the study. In the sentence “Prior work, such as Chia et al. (2024), has shown that VLMs often struggle at the perception stage of multimodal reasoning even before doing inductive and deductive reasoning,” please define each reasoning subcategory mentioned — namely perception, inductive, and deductive reasoning.

6) It would be helpful to discuss whether similar datasets already exist that provide step-level or perception-focused supervision for multimodal reasoning. Including a short comparison in the related work section—highlighting how your dataset differs from or improves upon prior multimodal process or perception reasoning dataset. Would strengthen the motivation and contextual grounding of the paper.

7) It appears that VisualPRM does not include the abstractive reasoning dataset used for VL-PRM training. It would be valuable to include an additional variant of VisualPRM trained with the abstractive reasoning dataset and compare its performance. Such an experiment could reveal whether the gains observed in VL-PRM stem partly from the inclusion of abstract reasoning data. This comparison would offer deeper insight into the contribution of dataset composition to reasoning generalization.

8) “MMMU, which requires expert-level subject knowledge”, could you elaborate on what subject knowledge is needed and how it is different from the external knowledge needed to answer mathvista?

9) The statement “This finding is noteworthy, as it suggests that process reward models (PRMs) can unlock latent reasoning capacity that is not apparent under standard greedy decoding” seems overly general. The results do not appear to consistently support this claim across all benchmarks — particularly MMMU, MathVista, and MathVision.

10) The setting in Table 3 is not clear to me. Could you please elaborate on the experimental setup? How is performance evaluated for the GPT models and the Qwen-2.5-VL variants (3B, 7B, 32B)? It appears that the PRM outperforms the GPT-based models; however, since step correctness is determined by the GPT model acting as the judge, how is this performance boost obtained?

11) The statement “Our experiments reveal that training on the VL-PRM300K-balanced subset does not lead to consistent improvements: in some cases, performance increases marginally, while in others, a slight decrease is observed” is unclear. What is the intended purpose of the balanced subset? Could you please elaborate on its motivation and how balancing affects the overall training or evaluation setup?

12) “One-shot Search shows improved performance across VL-PRMs and benchmarks. This method is similar to Outcome Reward Modeling (ORM). What is interesting is that, despite not training an explicit ORM, our PRM behaves like one.”
This presents an interesting perspective — does it suggest that a fine-grained ORM is more effective than a PRM? However, this seems to contradict the earlier motivation that step-wise supervision provides better guidance. Should this be interpreted as indicating that a reward given only at the end of the reasoning process is more beneficial?

**Questions:**

1) “First, a perception-based description of the image is produced, followed by inductive and deductive reasoning steps to solve the problem.” Is a score assigned to each reasoning stage here? What do s_1, s_2, s_3, ... correspond to in the mathematical notation for Guided Greedy Search, One-shot Search, and Step-Score Aggregation?

---

### Note · Authors · 2025-11-13

I have read and agree with the venue's withdrawal policy on behalf of myself and my co-authors.